# NS-Gym: A Comprehensive and Open-Source Simulation Framework for Non-Stationary Markov Decision Processes

**Nathaniel S Keplinger,**[*]
Department of Computer Science
Vanderbilt University
Nashville, USA
nathaniel.s.keplinger@vanderbilt.edu

**Baiting Luo,**
Department of Computer Science
Vanderbilt University
Nashville, USA
baiting.luo@vanderbilt.edu

**Yunuo Zhang,**
Department of Computer Science
Vanderbilt University
Nashville, USA
yunuo.zhang@vanderbilt.edu

**Kyle Hollins Wray,**
Khoury College of Computer Sciences
Northeastern University
Boston, MA
k.wray@northeastern.edu

**Aron Laszka,**
Information Sciences and Technology
Pennsylvania State University
State College, PA
aql5923@psu.edu

**Abhishek Dubey,**
Department of Computer Science
Vanderbilt University
Nashville, USA
abhishek.dubey@vanderbilt.edu

**Ayan Mukhopadhyay**[*]
Department of Computer Science
Vanderbilt University
Nashville, USA
ayan.mukhopadhyay@vanderbilt.edu

## Abstract

Many real-world applications require decision-making where the environmental dynamics evolve over time. These non-stationary environments pose significant challenges to traditional decision-making models, which typically assume stationary dynamics. Non-stationary Markov decision processes (NS-MDPs) offer a framework to model and solve decision problems under such changing conditions. However, there are no standardized simulation frameworks for NS-MDPs, as opposed to widely popular frameworks for stationary problems. We present NS-Gym, the first simulation toolkit designed explicitly for NS-MDPs, integrated within the popular Gymnasium framework. In NS-Gym, we segregate the evolution of the environmental parameters that characterize non-stationarity from the agent's decision-making module, allowing for modular and flexible adaptations to dynamic environments. We review prior work in this domain and present a toolkit encapsulating key problem characteristics and types in NS-MDPs. This toolkit is the first effort to develop a set of standardized interfaces and benchmark problems to enable consistent and reproducible evaluation of algorithms under non-stationary conditions. We also benchmark several algorithmic approaches

---

[*]Corresponding authors

39th Conference on Neural Information Processing Systems (NeurIPS 2025) Track on Datasets and Benchmarks.

from prior work on NS-MDPs using NS-Gym. We envision that NS-Gym will enable researchers to study decision-making under non-stationarity by providing standardized interfaces, simulation frameworks, and benchmark problems. Project documentation, webpage, and code can be found at `https://nsgym.io` and `https://github.com/scope-lab-vu/ns_gym`

# 1 Introduction

Many real-world problems involve agents making sequential decisions over time under exogenous sources of uncertainty. Such problems exist in autonomous driving [Kiran *et al.*, 2021], medical diagnosis and treatment [Yu *et al.*, 2021], emergency response [Mukhopadhyay *et al.*, 2022], vehicle routing [Li *et al.*, 2021], and financial portfolio optimization [Pendharkar and Cusatis, 2018]. We define an *agent* as an entity capable of computation that *acts* based on *observations* from the environment [Kochenderfer *et al.*, 2022]. Decision-making for such agents is widely modeled by Markov decision processes (MDPs), a general mathematical model for stochastic control processes.

A canonical challenge in such problems, motivated by practical scenarios, is non-stationarity, where the distribution of environmental conditions can change over time. While non-stationarity has been well-explored from both control and decision-theoretic perspectives, several conceptual paradigms of non-stationarity exist, which lead to different mathematical formalisms for how the environmental parameters change and how the agent interacts with the changes. Ackerson and Fu [1970] provide one of the earliest conceptualizations of a system operating in "switching" environments, where the mean and covariance of the underlying process can change over time. Campo *et al.* [1991] formalize the switching process, where some environment parameters can change after a random sojourn time, as a sojourn-time-dependent Markov chain, which is semi-Markovian.

Recent investigations of non-stationary stochastic control processes involve two major threads: the first problem deals with an agent trying to adapt to a single change in the environment (which can either be observed [Pettet *et al.*, 2024] or unobserved [Luo *et al.*, 2024]); and the second problem models situations where environmental parameters can change continuously over time [Lecarpentier and Rachelson, 2019]. In an orthogonal line of work, Chandak *et al.* [2020b] present a problem formulation where the agent's goal is to maximize a forecast of future performance (of the control policy) instead of directly modeling the non-stationarity. Notably, these problem classes provide fundamentally different formalisms (or treatments) for non-stationarity.

Indeed, not only are the formalisms different, but we point out another interesting observation from prior work on non-stationary stochastic control processes: while recent prior work on *stationary* Markov decision processes (MDP) use standard benchmark problems, e.g., by using the popular Gymnasium toolkit Towers *et al.* [2023], there are no standard problems or benchmarks for *non-stationary* MDPs. For example, Lecarpentier and Rachelson [2019] evaluate non-stationarity using a custom non-stationary bridge environment (an abstract problem where an agent must navigate on a grid-based slippery maze where the properties of the surface change over time), Chandak *et al.* [2020b] use problems motivated by real-world applications such as recommendation systems and diabetes treatment, and Pettet *et al.* [2024] use well-known benchmark problems used for stationary MDPs (e.g., the cartpole problem from Gymnasium [Towers *et al.*, 2023]) and introduce non-stationarity manually.

In this paper, we identify key characteristics that serve as desiderata for non-stationary MDPs, review prior work in this area, and present the first simulation toolkit specifically tailored for non-stationary MDPs. We argue that four key considerations affect decision-making in non-stationary MDPs, where environmental factors can change over time: *what* changes? *how* does it change? can the agent *detect* the change? can the agent *know* the updated parameter that has changed? These questions summarize the nature of the change and the key properties of modeling approaches from prior work. Based on these questions, we present NS-Gym (Non-Stationary Gym), the first collection of simulation environments for non-stationary MDPs.

Inspired by the seminal work of Campo *et al.* [1991], we segregate the evolution of the environmental parameters that characterize non-stationarity and the agent's decision-making module. This modularization enables us to configure various components (and types) of non-stationary MDPs seamlessly. The NS-Gym toolkit is based on Python and is completely compatible with the widely popular Gymnasium framework. Instead of developing a new simulation environment from scratch,

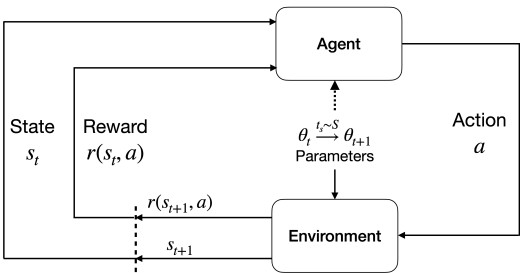

Figure 1: An overall framework for non-stationary Markov decision processes. At time $t$, the agent observes the state $s_t \in S$ and takes an action $a \in A$. The environment emits a reward signal $r(s_t, a)$ and transitions to the next state $s_{t+1}$. The transition and the reward are governed by parameters $\theta$, which do not necessarily have a stationary distribution. In general, the evolution of $\theta$ occurs through a semi-Markov chain whose *sojourn* time is distributed as $S$, which might be non-memoryless. Depending on the problem, the agent can detect and/or observe the evolution of $\theta$.

we build upon the existing Gymnasium toolkit due to its popularity and ensure that the large user base already familiar with Gymnasium can easily use NS-Gym (we keep all standard Gymnasium functionalities and interfaces intact). We make the following contributions:

**1)** We present the first simulation toolkit for NS-MDPs that provides a tailored, standardized, and principled set of interfaces for non-stationary environments.

**2)** We identify canonical problem instances for decision-making in non-stationary environments, e.g., decision-making where the agent *knows* about the change but is not *aware* of exactly what the change is, or decision-making where the agent is aware of the change.

**3)** We present an overview of prior work on non-stationary decision-theoretic models and a programming interface that unifies prior work.

**4)** Our simulation framework extends the widely popular Gymnasium toolkit, thereby requiring minimal added efforts from researchers in online planning, reinforcement learning, and decision-making in using our toolkit.

**5)** We present the first set of benchmark results (and open-source implementations using NS-Gym) that compares six algorithmic approaches for solving NS-MDPs.

**6)** Our benchmark results are presented across a series of problem types in non-stationary environments.

The rest of the paper is organized as follows. We begin by describing characteristics of NS-MDPs and prior work. Then, we identify canonical problem instances, describe our framework, and present a tutorial of how to use it. Finally, we present benchmark results using NS-Gym.

## 2 Characteristics of NS-MDPs and Prior Work

We begin by describing a comprehensive framework for decision-making in non-stationary environments. Admittedly, we point out that the conceptual boundaries of what constitutes an *agent* are unclear in this context. Instead, we leave this question open and point out the key components relevant to decision-making; whether these components are part of the agent or those supporting the agent is orthogonal to our discussion.

We refer to an agent as an entity that receives observations from an environment and can act or make decisions that interact with said environment. For simplicity, we assume a discrete-time process, although this discussion also extends to continuous-time stochastic control processes. Our fundamental model is that of a Markov decision process [Puterman, 2014]. We refer to the current state of the environment by $s \in \mathcal{S}$ and an action by $a \in \mathcal{A}$, where $\mathcal{S}$ and $\mathcal{A}$ denote the set of all states and actions, respectively. After taking an action: 1) the agent receives a scalar signal $r(s, a)$ from the environment, which can be perceived as a reward or a loss and is a measure of the agent's utility, and 2) the agent transitions to a new state, governed by a transition function $P(s' \mid s, a, \theta)$, where $\theta \in \Theta$

| Model | Reference | Is the change notified? | Is the change known? | What changes? | Nature of the change | Is the change bounded? |
|---|---|---|---|---|---|---|
| Piecewise Stationary MAB | Garivier and Moulines [2011] | No | No | Reward | The reward distribution is fixed over certain time periods, and then changes at unknown time steps. | No |
| Non-stationary MAB | Besbes *et al.* [2014] | No | No | Reward | The reward can change at arbitrary time points. | Yes |
| Piecewise Stationary MDP | Auer *et al.* [2008] | No | No | Transition, Reward | Bounded change analyzed as part of the UCRL2 algorithm | Yes |
| Non-Stationary MDP | Cheung *et al.* [2020] | N/A | No | Transition, Reward | The reward and transition can change at every time step | Yes |
| Non-Stationary MDP | Chandak *et al.* [2020b] | Yes | No | Transition, Reward | Inter-episodic transition and reward change; unchanging within an episode | No |
| Non-Stationary MDP | Chandak *et al.* [2020a] | Yes | No | Transition, Reward | Inter-episodic transition and reward change; unchanging within an episode | Yes |
| Non-Stationary MDP | Lecarpentier and Rachelson [2019] | Yes | Yes | Transition | The agent knows the current parameters, but not the future evolution. | Yes |
| Non-Stationary MDP | Pettet *et al.* [2024] | Yes | Yes | Transition | A single discrete change | Yes |
| Non-Stationary MDP | Luo *et al.* [2024] | Yes | No | Transition | A single discrete change | No |
| Non-Stationary Bandits with Periodic Variation | Chakraborty and Shettiwar [2024] | No | No | Reward | Periodic Variation | Yes |
| Time-Varying MDP | Ornik and Topcu [2021] | Yes | No | Transition | An unknown transition function must be learned. | Yes |

Table 1: Prior work on non-stationary Markov decision processes, categorized by important characteristics that affect decision making.

denotes a set of observable environmental parameters. We argue that explicitly specifying $\theta$ is critical to modeling non-stationary decision-making problems, as highlighted below.

We show a schematic of the major decision-theoretic components in Figure 1. In a non-stationary stochastic control process, the environmental parameters $\theta$ or the agent's utility function $r(s, a)$ can change over time. The manner in which the change evolves over time can be modeled by a Markov chain or, more generally, by a semi-Markov chain as proposed by Campo *et al.* [1991]. While this formalism has often not been used in recent work (which has focused less on the statistical properties of the changes), we argue that a formal representation of how the environmental parameters evolve is particularly important from the perspective of studying NS-MDPs. We use the same high-level formalism as Campo *et al.* [1991], i.e., the parameters $\theta$ evolve in time through a sojourn time distribution, which can be non-memoryless, thereby making the resulting stochastic process semi-Markovian [Hu and Yue, 2007]. If the sojourn-time distribution is memoryless, then the resulting process is a continuous-time Markov chain [Hu and Yue, 2007].

Motivated by how decision-making components are implemented in practice, we introduce two additional components: first, we introduce a runtime monitor that tracks the parameters $\theta$ and *detects* changes; in practice, the monitor can be implemented as an anomaly detector [Chandola *et al.*, 2009]. Note that while a monitor can track and detect changes in $\theta$; it might not be equipped to update the transition model $P$. From the agent's perspective, we refer to the ability to detect these environmental changes as *receiving a notification* about the change; note that we use this terminology to emphasize the segregation between the agent and the anomaly detector.

We introduce a second component, a *model updater*, which is a computational entity that can update the transition model by observing the changed parameters $\theta$. We do not argue that every agent designed for decision-making in non-stationary environments must have these components; indeed, we point out algorithmic prior work where one or both of these components are absent. Instead, we argue that these components sufficiently describe the infrastructure required for decision-making in non-stationary environments, whether a specific agent designs these components or simply assumes their existence is orthogonal to our discussion. Given these components, we categorize prior work in non-stationary stochastic control processes by answering four key questions in Table 1.

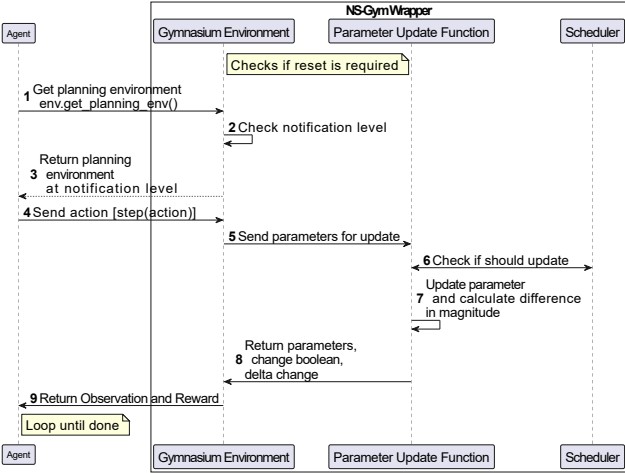

Figure 2: A sequence diagram of the agent-environment interaction in NS-Gym. Steps 4–9 in the diagram show how parameters are updated. Step 6 checks the current MDP time step and notifies if the parameter should be updated. Step 9 returns Observation and Reward types.

# 3   Framework Description

In this section, we outline the general structure of NS-Gym, elaborate upon our design decisions, and describe the general experimental pipeline using NS-Gym. The project's source code can be found at https://github.com/scope-lab-vu/ns_gym. We develop NS-Gym to allow researchers access to the breadth of NS-MDP specifications in the literature while maintaining the familiar interface popularized by the Gymnasium library [Towers *et al.*, 2023]. In Gymnasium, the *environment* object represents an MDP, defining states, actions, transitions, and rewards. The *observation* object provides the current state information available to the agent. The *Info* dictionary offers auxiliary diagnostic data, useful for debugging but inaccessible by the agent. The standard Gymnasium workflow involves initializing the environment, then looping: the agent observes, selects an action, executes it, and receives the next observation, reward, and done status. Once an episode ends, the environment is reset.

NS-Gym provides a set of wrappers to augment the classic control suite of Gymnasium environments, three grid world environments, and the MuJoCo [Todorov *et al.*, 2012] control environment. These wrappers modify exposed parameters of the base environments (the stationary counterparts of our non-stationary environments) to introduce non-stationarity, either at each decision epoch or via user-defined functions. For example, in a deterministic environment such as "CartPole", an example change is varying the value of the gravity, thereby altering the dynamics of the cart. In stochastic environments, the probability distribution over possible next states, given the current state action pair, changes. For example, in the Frozen Lake environment, this change might increase (or decrease) the coefficient of friction, making the movement of the agent more (or less) uncertain. Figure 2 illustrates the high-level structure of the wrapper and Table 2 lists the NS-Gym environments and their observable parameters, with detailed descriptions in the appendix. Modifying these parameters directly changes the NS-MDP's core transition function.

A key feature of NS-Gym is its management of the agent-environment interaction during training, planning, and testing. For rigorous evaluation of policies on NS-MDPs the decision-making algorithm must *not* have future ground-truth knowledge of how an NS-MDP may evolve. During training and planning time the decision-making agent executes actions on and receives observations from a stationary version of the NS-Gym environment object. While the agent interacts with a stationary version of the NS-Gym environment during planning, at test time, the execution of all actions and received observations occurs on a time-dependent NS-MDP, even if planning uses a static model.

These agent-environment interactions encapsulate the following *problem types*, which we explain using the Frozen Lake environment. Consider a Frozen Lake environment where the agent's probability

of going in its intended direction is $\theta_1$ in the base environment. Now, the lake becomes more slippery, and this probability changes to $\theta_2$. We model the following key problem types:

**1)** The agent receives a message that the extent to which the lake is slippery has changed (corresponding to a successful anomaly detection), but it is unaware of the exact change (i.e., it does not know $\theta_2$). This setting is motivated by prior work by Luo *et al.* [2024]).

**2)** The user is aware of the exact environmental change, i.e., it knows $\theta_2$ but in non-stationary settings, the agent might not have time to train a new policy. This setting is motivated by prior work by Pettet *et al.* [2024] and Lecarpentier and Rachelson [2019].

**3)** Problems where the agent is not notified about the change, i.e., it is unaware that the probability is no longer $\theta_1$. This setting is motivated by prior work by Garivier and Moulines [2011].

**4)** In an orthogonal thread, we identify the frequency of the change, i.e., problems with a single change in an environment variable [Luo *et al.*, 2024; Pettet *et al.*, 2024] (e.g., the change is from $\theta_1$ to $\theta_2$) or multiple changes within an episode [Cheung *et al.*, 2020] (e.g., the change is $\theta_1 \to \theta_2 \to \theta_3 \to \ldots$) or changes within multiple episodes [Chandak *et al.*, 2020b].

Users can configure notifications the agent receives about changes in the NS-MDP at three distinct levels: **1) Basic Notification:** The agent receives a Boolean flag indicating a change in an environment parameter, **2 ) Detailed Notification:** In addition to the Boolean flag, the agent is informed of the magnitude of the change and **3 ) Full Environment Model:** If the agent requires an environmental model for planning purposes (such as in Monte Carlo tree search), NS-Gym can provide a stationary snapshot of the environment. This snapshot aligns with the basic or detailed notification settings configured by the user. If the user seeks a model without detailed notification, the planning environment is a stationary snapshot of the base environment. Conversely, if detailed notifications are enabled, the agent receives the most up-to-date version of the environment model (but not any future evolutions).

To accommodate information unique to non-stationary environments, NS-Gym uses custom observation and reward data types. The custom observation type has four fields: `state`, `env_change`, `delta_change`, and `relative_time`. The `state` field encodes the current environment state. The `env_change` field is a dictionary of Boolean flags indicating what environment parameter has changed. The `delta_change` reports the amount of change in each environment parameter. By default, NS-Gym returns the difference in value for scalar parameters and the Wasserstein distance for probability distributions. The `relative_time` is the number of decision epochs that have lapsed since the episode's start. The reward type is similarly constructed, but instead of the `state` field, we have a `reward`.

NS-Gym's API is designed to handle a diverse range of custom non-stationary MDPs, allowing users to flexibly model dynamic environments through configurable "schedulers" and "parameter" update functions. We decouple the timing (and thereby, the frequency) and the manner of parameter changes, providing users with greater flexibility.

Schedulers are functions that determine when environmental changes *should* occur by returning a Boolean flag at each time step. If a scheduler returns `True`, the update functions modify the specified parameter accordingly. NS-Gym includes built-in schedulers for continuous, stepwise, random, and periodic time steps with support for custom schedulers via subclassing. Update functions define *how* parameters change, with examples like random walks with budget-bounded constraints or bounded by Lipschitz continuity [Lecarpentier and Rachelson, 2019].

Mathematically, at a high level NS-Gym defines:

1. A transition function $P(s' \mid s, a, \theta)$ that defines the probability of transitioning to state $s'$ given the current state, action taken, and set of environmental parameters, $\theta$.

2. A function $f(t)$ i.e $\theta_{t+1} = f(\theta_t, t)$ that controls how $\theta$ evolves over time.

In our framework, we implement $f(t)$ as "schedulers", which determine when updates occur, and "parameter update functions" which determine how $\theta$ changes.

## 3.1 Experimental Pipeline

The general NS-Gym experimental setup procedure is: 1) *Create a Standard Gymnasium Environment*. 2) *Define Parameters to Update*. 3) *Map Parameters to Schedulers and Update Functions*. 4) *Generate a Non-Stationary Environment*. Consider that a user needs to model a non-stationary CartPole environment with increasing pole mass (0.1/step), random walk gravity (every 3 steps), and basic agent notifications, the following NS-Gym code illustrates the setup:

The first step involves importing ns_gym, i.e.,

```
import gymnasium as gym
import ns_gym
```

Next, we create the base gymnasium environment, i.e.,

```
env = gym.make("CartPole-v1")
```

Next, to describe the evolution of the non-stationary parameters, we define the two schedulers and update functions that model the semi-Markov chain over the relevant parameters, i.e.

```
scheduler_1 = ns_gym.schedulers.ContinuousScheduler()
scheduler_2 = ns_gym.schedulers.PeriodicScheduler(period = 3)
U_Fn_1 = ns_gym.update_functions.IncrementUpdate(scheduler_1,k = 0.1)
U_Fn_2 = ns_gym.update_functions.RandomWalk(scheduler_2)
```

Next, we map the parameters to the update functions, i.e.,

```
tunable_params = {"masspole":U_Fn_1,"gravity": U_Fn_2}
```

Then, we set the notification level and pass the parameters and environment into the wrapper, i.e.

```
ns_env = ns_gym.wrappers.NSClassicControlWrapper(env, tunable_params,
    change_notification=True)
obs,info = ns_env.reset()
```

Finally, we grab an environment model for planning, i.e.,

```
planning_env = ns_env.get_planning_env()
```

NS-Gym enables efficient experimentation via Gymnasium's parallelization and vectorization API. Appendix D provides API examples and execution time measurements. We also provide a detailed user tutorial in the supplementary material.

| Environment Name | Tunable Environmental Parameters |
|---|---|
| **Acrobot** | dt, LINK_LENGTH_1,LINK_LENGTH_2, LINK_MASS_1, LINK_MASS_2, LINK_COM_POS_1, LINK_COM_POS_2, LINK_MOI |
| **CartPole** | gravity, masscart, masspole, force_mag, tau, length |
| **MountainCar** | gravity, force |
| **Continuous_MountainCarEnv** | power |
| **Pendulum** | m, l, dt, g |
| **FrozenLake** | P |
| **CliffWalking** | P |
| **Bridge** | P, left_side_prob, right_side_prob |

Table 2: Environmental parameters for classic control and gridworld environments as they appear in NS-Gym. See Appendix G for details.

## 3.2 Evaluating Non-Stationary Markov Decision Processes

Evaluating the difficulty or nature of the uncertainties of NS-MDPs is often imperative for decision-making; however, it is challenging to simulate directly in practice. NS-Gym's *eval* module offers methods to assess non-stationary MDPs. Ideally, as an NS-MDP evolves from an initial MDP, $M_0$, to a subsequent MDP, $M_1$, we could compute the *regret* of any arbitrary policy $\pi$ with respect to an optimal policy for $M_1$, $\pi_1^*$, i.e.,

$$\text{Regret} = \mathbb{E}_{p_1} \sum_{t=0}^{\infty} \gamma^t r_{M_1}(s_t, \pi_1^*(s_t)) - \mathbb{E}_{p_1} \sum_{t=0}^{\infty} \gamma^t r_{M_1}(s_t, \pi_0^*(s_t))$$

However, $\pi_1^*$ is typically unknown.[1] Instead, NS-Gym includes two baseline evaluation functions for assessing the complexity of an NS-MDP. The first function is an ensemble performance metric that computes the mean regret across $N$ episodes and a set of *stable* policies in $M_0$, denoted by $\Pi_s(M_0)$, i.e.,

$$\text{Ensemble Regret} = \frac{1}{|\Pi_s(M_0)| \cdot N} \Big\{ \sum_{\pi \in \Pi_s(M_0)} \sum_{n=1}^{N} \sum_{(s,a) \in \tau_n(\pi, M_1)} r_{M_1}(s, \pi(s))$$
$$- \sum_{\pi \in \Pi_s(M_0)} \sum_{n=1}^{N} \sum_{(s,a) \in \tau_n(\pi, M_0)} r_{M_0}(s, \pi(s)) \Big\}$$

where $\tau_n(\pi, M)$ denotes the $n$-th trajectory by following an arbitrary policy $\pi$ in an MDP $M$. Users can supply their own policies or leverage the NS-Gym interface with Stable-Baselines3 [Raffin *et al.*, 2021] to access reinforcement learning models across Gymnasium environments. For convenience, we provide a set of pre-trained model weights from Stable-Baselines3, along with additional built-in algorithms.

For stochastic environments, we also provide a policy-agnostic metric, the PAMCTS-Bound [Pettet *et al.*, 2024], which measures the maximum difference in transition functions between two MDPs, $M_0$ and $M_1$, and is defined as:

$$\forall s, a : \text{PAMCTS-Bound} = ||P(s,a)_{M_0} - P(s,a)_{M_1}||_\infty$$

where $s$ denotes environment states and $a$ denotes actions. Custom NS-MDP evaluation metrics can be easily added by inheriting from NS-Gym's base "Evaluator" class, which takes an NS-Gym environment object and returns the computed metric.

## 4 Benchmark Experiments

In this section, we demonstrate the utility of NS-Gym by evaluating decision-making algorithms in environments built using the library. We explore the following questions: How well can an agent adapt when environmental change is known or unknown? What if the system undergoes continuous evolution? How well can an agent handle frequent updates? Additionally, we show how NS-Gym can simulate a related modeling paradigm, the contextual Markov decision process.

We benchmark six algorithms across four base environments, namely CartPole, FrozenLake, CliffWalker, and Bridge environments. Additional results for other environments are in the supplemental material. We evaluate both discrete and continuous transition changes, testing each environment-agent pair with no notification, basic notification, or an up-to-date model. Performance is measured using cumulative undiscounted episodic rewards.

For CartPole, we vary the pole's mass in single and continuous experiments. In the grid-world environments, we adjust the probability of moving in the intended direction, with single experiments shifting from a default value to $0.4$, $0.6$, or $0.8$, and continuous experiments decreasing the probability incrementally until a threshold is reached. Additional details on environment setup and experimental results for the Pendulum, Mountain Car, and Acrobot are in the supplemental material. Note that planning and training for all benchmark algorithms are done in stationary versions of the environment. Test time execution happens in the time-varying NS-MDPs.

We evaluate the non-stationary environment across six different decision-making agents: Monte Carlo tree search (MCTS) [Kocsis and Szepesvári, 2006], double deep Q learning (DDQN) [van Hasselt *et al.*, 2015], AlphaZero [Silver *et al.*, 2017], adaptive Monte Carlo tree search (ADA-MCTS) [Luo *et al.*, 2024], risk-averse tree search (RATS) [Lecarpentier and Rachelson, 2019], and policy-augmented Monte Carlo tree search (PAMCTS)[Pettet *et al.*, 2024]. Note that our work is the *first effort to benchmark approaches for tackling non-stationarity on standardized problem settings*. See Appendix I for algorithm details. For all environments, model-based algorithms are provided with a stationary snapshot of the model at the appropriate notification level.

A closely related, but *somewhat* different modeling paradigm involves contextual Markov decision processes (C-MDPs). C-MDPs extend standard MDPs by introducing a "context" space, where

---

[1]Note that if $\pi_1^*$ is known, users can easily use NS-Gym to compute the regret.

each context parameterizes a specific variation within a family of related MDPs [Hallak *et al.*, 2015]; intuitively, a contextual MDP can be thought of as a special case of an NS-MDP which *only* exhibits inter-episodic non-stationarity (defined by the context). As a result, C-MDPs can be easily implemented using the NS-Gym tool. Similar to its approach with NS-MDPs, NS-Gym allows for setting a new value for an environmental parameter between episodes, establishing a new context. NS-Gym then provides an interface to evaluate policies across a range of contexts of a C-MDP. To demonstrate this capability, we show how NS-Gym can evaluate a full contextual MDP, enabling experiments that study transfer learning. Table 5 shows the generalized performance results for the sequential task selection problem on two simple task selection baselines, random task selection and model-based transfer learning with equidistant task selection [Cho *et al.*, 2025]. Algorithmic details and C-MDP background information are found in the supplemental material.

| | | MCTS | AlphaZero | DDQN | PAMCTS 0.25 | PAMCTS 0.5 | PAMCTS 0.75 | ADA-MCTS | RATS |
|---|---|---|---|---|---|---|---|---|---|
| **Bridge** | 0.4 | -0.58 ± 0.47 | -0.26 ± 0.56 | -0.82 ± 0.33 | -0.58 ± 0.27 | -0.20 ± 0.33 | **-0.16 ± 0.33** | -0.54 ± 0.07 | -0.98 ± 0.02 |
| | 0.6 | -0.18 ± 0.57 | **0.58 ± 0.47** | -0.78 ± 0.36 | 0.46±0.33 | 0.46 ± 0.3 | 0.38 ± 0.31 | -0.16 ± 0.09 | 0.05 ± 0.08 |
| | 0.8 | 0.64 ± 0.45 | **0.92 ± 0.23** | -0.72 ± 0.4 | 0.4 ± 0.31 | 0.72 ± 0.23 | 0.8 ± 0.2 | 0.46 ± 0.09 | -0.01 ± 0.01 |
| **Frozen Lake** | 0.4 | 0.11 ± 0.18 | 0.06 ± 0.02 | 0.22 ± 0.17 | 0.15 ± 0.04 | 0.16 ± 0.03 | 0.12 ± 0.03 | **0.67 ± 0.05** | 0.6 ± 0.05 |
| | 0.6 | 0.25 ± 0.25 | 0.25 ± 0.04 | 0.66 ± 0.19 | 0.3 ± 0.05 | 0.33 ± 0.05 | 0.27 ± 0.04 | 0.56± 0.05 | **0.88 ± 0.03** |
| | 0.8 | 0.53 ± 0.29 | 0.39 ± 0.05 | 0.91 ± 0.12 | 0.74 ± 0.04 | 0.68 ± 0.05 | 0.54 ± 0.05 | 0.49 ± 0.05 | **0.97 ± 0.02** |
| **Cliff Walking** | 0.4 | -1593.89 ± 68.9 | -543.94 ± 45.98 | -1742.54 ± 91.29 | -1572.21 ± 60.82 | **-477.50 ± 54.66** | -1382.04 ± 77.88 | -1503.34 ± 53.57 | -777.55 +/- 31.19 |
| | 0.6 | -1216.72 ± 63.68 | **6.97 ± 8.2** | -1018.27 ± 96.95 | -1159.77 ± 53.85 | -374.64 ± 44.31 | -477.50 ± 54.65 | -1019.72 ± 35.99 | -314.84 ± 12.8 |
| | 0.8 | -773.62 ± 54.67 | **64.41 ± 3.44** | -287.17 ± 40.55 | -790.60 ± 46.66 | -54.22 ± 14.25 | -109.08 ± 25.99 | -523.73 ± 23.79 | -231.86 ± 4.22 |
| **Cart Pole** | 1 | **600.90 ± 47.68** | 441.1 ± 51.96 | 135.53 ± 0.28 | 525.98 ± 31.91 | 120.48 ± 0.57 | 135.41 ± 0.32 | – | – |
| | 1.5 | **641.28 ± 50.47** | 272.82 ± 21.25 | 139.19 ± 0.27 | 467.35 ± 25.11 | 117.60 ± 1.24 | 135.42 ± 0.34 | – | – |

Table 3: Mean episode reward with standard error with a single exogenous change without notification (see supplementary material for results with notification). The best-performing agents are in bold. Blanks denote settings where the algorithm is not applicable.

| | | MCTS | AlphaZero | DDQN | PAMCTS 0.25 | PAMCTS 0.5 | PAMCTS 0.75 | ADA-MCTS | RATS |
|---|---|---|---|---|---|---|---|---|---|
| **Bridge** | WN | 0.18 ± 0.1 | **0.6 ± 0.08** | -0.44 +- 0.09 | 0.28 ± 0.56 | 0.34 ± 0.54 | 0.08 +/ 0.56 | – | 0.36 ± 0.09 |
| | WON | 0.04 ± 0.10 | **1.00 ± 0.00** | -0.84 ± 0.05 | -0.02 ± 0.58 | 0.22 ± 0.57 | 0.20 ± 0.57 | 0.08 ± 0.1 | 0.36 ± 0.09 |
| **Frozen Lake** | WN | 0.15 ± 0.04 | 0.25 ± 0.04 | 0.1 ± .04 | 0.2 ± 0.04 | 0.15 ± 0.04 | 0.04 ± 0.02 | – | **0.71 ± 0.05** |
| | WON | 0.24 ± 0.04 | 0.25 ± 0.04 | 0.27 ± 0.04 | 0.14 ± 0.03 | 0.21 ± 0.04 | 0.08 ± 0.03 | 0.59 ± 0.05 | **0.71 ± 0.05** |
| **Cliff Walking** | WN | -847.48 ± 55.83 | **77.95 ± 0.40** | -137.89 ± 29.19 | -803.94 ± 54.89 | -56.56 ± 19.2 | -75.06 ± 20.77 | – | -932.89 ± 50.55 |
| | WON | -907.67 ± 54.62 | **76.0 ± 1.89** | -359.97 ± 42.46 | -732.28 ± 53.50 | -31.84 ± 14.97 | -132.26 ± 26.98 | -1144.91 ± 43.83 | -707.65 ± 36.33 |
| **Cart Pole** | WN | 702.7 ± 21.95 | 203.68 ± 1.35 | 100.78 ± 2.62 | **1392.23 ± 65.57** | 96.15 ± 2.5 | 99.95 ± 2.58 | – | – |
| | WON | 149.0 ± 1.79 | **251.47 ± 5.81** | 95.97 ± 2.68 | 109.39 ± 2.69 | 55.17 ± 1.7 | 95.61± 2.73 | – | – |

Table 4: Mean episode reward with standard error with continuous parameter updates. WN and WON denote settings "with notification" and "without notification" respectively. The best-performing approaches are in bold. Blanks denote settings where the algorithm is not applicable.

| **Environment:** `Observable Parameter` | Random | MBTL-ES |
|---|---|---|
| **CartPole:** `masscart` | $0.67 \pm 0.03$ | $0.82 \pm 0.0175$ |
| **Acrobot:** `link mass 1` | $0.83 \pm 0.01$ | $0.82 \pm 0.01$ |
| **MountainCar:** `gravity` | $0.34 \pm 0.03$ | $0.19 \pm 0.03$ |
| **Pendulum:** `mass` | $0.74 \pm 0.01$ | $0.80 \pm 0.01$ |

Table 5: Normalized generalized performance of simple baselines for solving the sequential task selection problem ($k = 9$ tasks) on C-MDPs evaluated using NS-Gym.

Table 3 shows results from the single change experiments without notifications, and Table 4 reports agent performance in the continuous experiment setting with and without notification. We provide a complete table of experimental results and figures in the supplemental materials. From the benchmark results, we have derived some key insights about how different strategies perform under varying conditions. This analysis provides a clearer understanding of how algorithms respond to dynamic environmental changes.

**Impact of Detailed Notification on Performance with Single Transition Change**: The presence of detailed notifications generally enhances the performance of most methods. AlphaZero, MCTS, PA-MCTS, and RATS demonstrate marked improvements when notifications are available in some environments, effectively leveraging the most up-to-date dynamics to optimize decision-making processes. In contrast, DDQN shows only a modest improvement as it is difficult to adapt to changes in a limited time.

**Impact of Notification on Performance with continuous Transition Change**: The presence of detailed notifications generally improves the performance of most methods across various environments. This highlights the importance of quickly adapting the planning model to the latest dynamics of the environment. For example, methods like MCTS and PAMCTS, which leverage online search, show a consistent performance increase across different environments, emphasizing the effectiveness of an online approach in maintaining robust performance amid continuous changes when notifications are given. We observe that AlphaZero performs exceptionally well with notifications.

**Variability in Algorithm Effectiveness**: When comparing methods that incorporate risk-averse strategies with those that do not, it is evident that the ones with risk-averse strategies perform differently. In environments like FrozenLake, where the agent is more vulnerable to varying levels of unpredictability compared to other environments, methods like ADA-MCTS and RATS, which incorporate risk-averse strategies, generally perform better with single transition changes and continuous changes. These methods are designed to account for and mitigate the risks brought on by the environment's stochastic nature, leveraging worst-case sampling strategies to make decisions robust to possible changes. This enables them to navigate more effectively and avoid the pitfalls that non-risk-averse methods might encounter. We also point out that in prior work, ADA-MCTS is the only approach that can *learn*

## 5 Conclusion

We present NS-Gym, the first simulation toolkit and set of standardized problem instances and interfaces explicitly designed for NS-MDPs. NS-Gym incorporates problem types and features from over fifty years of research in non-stationary decision-making. We also present benchmark results using prior work. We will continue to maintain NS-Gym, extend it, and maintain a leaderboard of approaches.

## 6 Acknowledgments

This material is based upon work supported by the National Science Foundation (NSF) under Grants CNS-2238815 and CNS-2531369, and by the Defense Advanced Research Projects Agency (DARPA) and US Air Force Research Lab (AFRL) under the Assured Neuro Symbolic Learning and Reasoning program. Results presented in this paper were obtained using the Chameleon testbed supported by the National Science Foundation. Any opinions, findings, conclusions, or recommendations expressed in this material are those of the authors and do not necessarily reflect the views of the NSF, DARPA, or AFRL.

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

## A  Additional NS-Gym Framework Description

NS-Gym provides a simple set of wrappers built on top of the Gymnasium API. Table 6 shows an example of the custom NS-Gym observation type.

| Field Name | Data Type |
|---|---|
| state | Union[array,int] |
| env_change | Union[dict[str,bool],None] |
| delta_change | Union[dict[str,float],None] |
| relative_time | Union[int,float] |

**Observation Type**

Table 6: The custom observation types of NS-Gym capture essential components of NS-MDPs.

## B  Limitations and Broader Impact

There are limitations in the diversity of the base environments included in the NS-Gym package. We aim to broaden its scope to more complex tasks in the future. Beyond NS-Gym's core functionality, future efforts will focus on optimizing its environments and enhancing model update mechanisms to achieve improved runtimes and more efficient experimentation.

NS-Gym, as a toolkit, may not have an immediate societal impact. However, we anticipate that NS-Gym will enable community-driven development of algorithms for decision-making in non-stationary environments. These resulting algorithms hold the potential for substantial societal impact by tackling open problems in healthcare, transportation, and autonomous driving.

## C  NS-Gym Evaluation Module

NS-Gym includes an evaluation module designed to assess the "difficulty" of a custom NS-MDP. The `evaluation` module primarily consists of two submodules. The first submodule comprises the `run_experiment` helper functions, which execute a sequence of episode traces and log the results based on a provided agent, YAML configuration file, and NS-Gym environment. The second submodule focuses on `metrics`, a suite of evaluation functions as described in Section 3.2.

Below, we provide an example workflow for using the NS-Gym API to evaluate the difficulty of a custom NS-MDP. Consider designing a non-stationary version of the "Pendulum" environment, where at each MDP time step, the mass of the pendulum increases by 0.01 units. This environment can be constructed as follows:

```
import gymnasium as gym
from ns_gym.wrappers import *
from ns_gym.schedulers import *
from ns_gym.update_functions import *

env = gym.make("Pendulum-v1")
tunable_param = "m"
scheduler = ContinuousScheduler()
update_fn = IncrementUpdate(scheduler,0.01)
param_map  = {tunable_param:update_fn}
ns_env = NSClassicControlWrapper(env,param_map)
```

To evaluate the difficulty of this NS-MDP, we can test it using the `EnsembleMetric` evaluator, which computes the ensemble regret across a suite of policies and outputs the results to standard output:

```
from ns_gym.evaluate.metrics import EnsembleMetric
evaluator = EnsembleMetric()
ensemble_reward, perfomance_dict = evaluator.evaluate(ns_env,100)
```

An example of the standard output is shown below:

```
========================================
Evaluation Results
========================================
Ensemble Regret: -880.2925733559728

Agent Regret:
  - DDPG: -176.39195063398657
  - PPO: -561.0702939479152
  - A2C: -1589.8067051120454
  - TD3: -1193.9013437299438
========================================
```

## D  Execution Time, Vectorization, and Parallelization

In this section, we show runtime comparisons between NS-Gym and the base Gymnasium environments. Table 7 compares the execution time for a single step in the environment object *with* parameter updates. Naturally, how environmental parameters are updated between decision epochs will induce additional overhead depending on the complexity of the update operation. The tabular environments like CliffWalking and FrozenLake rebuild a complete state transition table whenever there is an update. If there is no parameter update the absolute overhead is minimal. The execution time to take a stationary snapshot of the environment for planning is shown in Table 8.

| Environment | Gymnasium ($\mu s$) | NS-Gym ($\mu s$) | Absolute Overhead ($\mu s$) |
|---|---|---|---|
| FrozenLake | $10.82 \pm 0.01$ | $299.47 \pm 0.07$ | $288.65 \pm 0.07$ |
| CliffWalking | $10.13 \pm 0.01$ | $124.50 \pm 0.07$ | $114.37 \pm 0.07$ |
| Acrobot | $90.90 \pm 0.03$ | $107.11 \pm 0.02$ | $16.21 \pm 0.03$ |
| MountainCar | $17.22 \pm 0.01$ | $29.85 \pm 0.01$ | $12.63 \pm 0.02$ |
| CartPole | $14.11 \pm 0.01$ | $28.11 \pm 0.01$ | $14.00 \pm 0.02$ |
| Pendulum | $32.34 \pm 0.01$ | $47.45 \pm 0.02$ | $15.11 \pm 0.02$ |

Table 7: Comparison of execution time for a single step in base Gymnasium environments and NS-Gym. The NS-Gym shows the execution time when there is a parameter update every timestep.

| Environment | env.get_planning_env() ($\mu s$) |
|---|---|
| FrozenLake | $2032.50 \pm 0.36$ |
| CliffWalking | $21393.41 \pm 60.26$ |
| Acrobot | $524.44 \pm 0.08$ |
| MountainCar | $491.35 \pm 0.08$ |
| CartPole | $573.05 \pm 0.08$ |
| Pendulum | $586.54 \pm 0.09$ |

Table 8: NS-Gym get updated planning environment execution time.

Built on Gymnasium, NS-Gym is compatible with its vectorization and parallelization APIs. Below is an example of how a user could set up an experiment with a vectorized environment with parallel execution. To vectorize the NS-Gym environments, the observation and reward types need to be modified into a dictionary of vectorizable objects. The `FlattenObsWrapper` and `FlattenRewardWrapper` do this.

```python
def make_env():
    def _init():
        env = gym.make('FrozenLake-v1', render_mode="rgb_array",
    is_slippery=False)
        scheduler = ContinuousScheduler()
        update_function = DistributionDecrementUpdate(scheduler=
    scheduler, k=0.1)
        param = "P"
        params = {param: update_function}
        ns_env = NSFrozenLakeWrapper(env, params, initial_prob_dist
    =[1, 0, 0],delta_change_notification=True, change_notification=
    True)
        ns_env = FlattenObsWrapper(ns_env)
        ns_env = FlattenRewardWrapper(ns_env)
        return ns_env
    return _init

vec_env = AsyncVectorEnv([make_env() for _ in range(4)])
out = vec_env.reset()
out = vec_env.step(actions=np.array([0 for x in range(4)]))
```

Table 9 shows environment steps per second for each environment running eight parallel asynchronous environments.

| Environment | Gymnasium (SPS) | NS-Gym (SPS) |
|---|---|---|
| FrozenLake | $83,978.50 \pm 626.08$ | $22,917.84 \pm 567.25$ |
| CliffWalking | $83,978.50 \pm 626.08$ | $22,917.84 \pm 567.25$ |
| Acrobot | $39,308.93 \pm 1,214.93$ | $29,803.10 \pm 965.21$ |
| MountainCar | $82,620.94 \pm 898.42$ | $42,802.29 \pm 2,026.04$ |
| CartPole | $83,421.50 \pm 770.88$ | $17,768.18 \pm 428.61$ |
| Pendulum | $54,259.26 \pm 216.82$ | $31,677.50 \pm 1,493.39$ |

Table 9: Parallel vectorized asynchronous execution for gridworld and classic control environments. For each environment, we run eight environment instances in parallel and we report environment steps per second (SPS).

## E   Additional Details on Simulating C-MDPs in NS-Gym

Contextual Markov decision processes (C-MDPs) are a class of MDP where variations in the underlying process create related but distinct environments [Hallak *et al.*, 2015]. Formally, a C-MDP is defined as $(\mathcal{C}, \mathcal{S}, \mathcal{A}, \mathcal{M}(c))$, where $\mathcal{C}$ is the context space, $\mathcal{S}$ is the state space, $\mathcal{A}$ is the action space, and $\mathcal{M}(c) = (\mathcal{S}, \mathcal{A}, P_c(s'|s,a), R_c(s,a))$ maps a context to a specific MDP instance. In this paper, we evaluate two simple baselines for the sequential source task selection problem, which aims to dynamically choose a subset of tasks from the entire context space to maximize expected performance.

We benchmark the following simple strategies for selecting training tasks on a fixed set of hyperparameters:

- **Random Task Selection**: $k$ tasks are randomly chosen from the complete context space.
- **Equidistant Task Selection**: $k$ tasks are selected such that they are uniformly spaced across the full context space.

For our experiments, we consider the following environments:

- **CartPole**: The context parameter is the cart's mass, defined over the range $[0.1, 10.0]$ units. A DQN model from StableBaselines-3 is trained for each context, using default hyperparameters.
- **Acrobot**: We vary the mass of the "first link" within the range $[0.1, 10.0]$ units. A PPO model from StableBaselines-3 is trained for each context with default hyperparameters.
- **MountainCar**: The context is gravity, ranging from $[0.0015, 0.006]$ units.
- **Pendulum**: The pendulum's mass is the varied context, ranging from $[1.0, 10.0]$ units. For this environment, a PPO model is trained using default StableBaselines-3 hyperparameters.

# F Hardware Specifications

The benchmarking experiments were conducted across a set of computing resources. Specifically, on one of the following machines:

- Server with single Intel Core i9-14900KS processor and single NVIDIA GeForce RTX 4090 GPU.
- Server with a single AMD Ryzen Threadripper 1950X 16-Core processor and single NVIDIA TITAN Xp GPU.
- Single processor Chameleon testbed machines.

The execution time experiments were all conducted on a single server with a Ryzen Threadripper 1950X 16-Core processor and a single NVIDIA TITAN Xp GPU. We executed all experiments in parallel using multiprocessing.

All experiments were executed in parallel using multiprocessing, with each experiment requiring at most approximately one day to complete.

# G Description of NS-Gym Environments

Below, we provide descriptions for each environment supported by NS-Gym. Table 2 outlines tunable parameters for classic control and gridworld environments as they appear in NS-Gym. Table 10 outlines tunable parameters for MuJoCo environments.

## G.1 CartPole

The CartPole environment has a discrete action space and a continuous state space. As illustrated in Figure 3, the agent's objective is to keep the pole balanced on top of the cart for as long as possible. The agent receives a reward of +1 for each time step that the pole remains balanced. The state is represented by a four-dimensional vector, which includes the cart's position, cart's velocity, pole's angle, and pole's angular velocity. At each time step, the agent can apply a fixed force to push the cart either left or right.

## G.2 Mountain Car

The MountainCar environment (see Figure 4) is a continuous state but discrete action space environment. In this environment, a car is stuck in a valley, and the agent must apply force to the cart to build momentum so that the car can escape. By default, the agent receives a zero reward for escaping the valley and a -1 reward otherwise. The agent can either push the car to the left, right, or not at all. The continuous Mountain Car environment is similar to the standard Mountain Car environment but with a continuous action space. In the continuous analog, the agent chooses the direction in which to apply the force to the car.

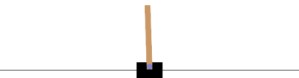

Figure 3: The Gymnasium CartPole environment.

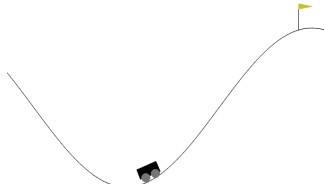

Figure 4: The Gymnasium MountainCar environment.

### G.3 Acrobot

The Acrobot environment is a double pendulum (see Figure 5). The agent can apply torque to the joint connecting the two links of the double pendulum to move the free end above a threshold height. At each time step, the agent can either apply +1, 0, or -1 units of torque.

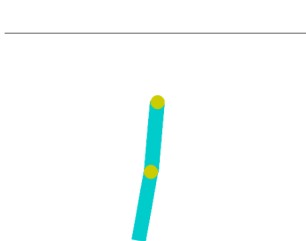

Figure 5: The Gymnasium Acrobot environment.

### G.4 Pendulum

The Pendulum environment is a continuous state and action space environment. The agent aims to keep the pendulum inverted for as long as possible. The agent receives a reward proportional to the pendulum's angle. At each time step, the agent applies some torque magnitude to the pendulum's free end. Figure 6 shows the pendulum environment.

### G.5 FrozenLake

The FrozenLake environment (Figure 7) is a stochastic, discrete action, and discrete state space grid-world environment. The agent navigates from a starting cell in the top left corner of the map to a jail cell in the bottom right corner while avoiding holes in the "frozen lake." The agent can move in an intended direction, with some probability that it will move in a perpendicular direction instead. The agent will get a reward of +1 if it reaches the goal and 0 otherwise.

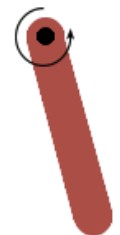

Figure 6: The Gymnasium Pendulum environment.

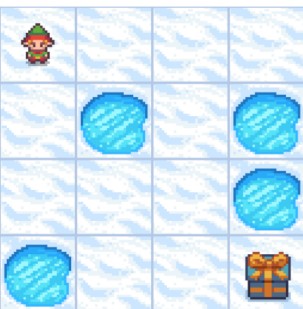

Figure 7: The Gymnasium FrozenLake environment.

## G.6 CliffWalking

The CliffWalking environment (Figure 8) is a deterministic grid-world environment. The agent must navigate from the start to the goal cell in the fewest steps. If the agent falls off a "cliff," it accrues a reward of -100 and resets at the start cell without ending the episode. The agent accrues -1 reward for each cell that is not a cliff or a goal state. The goal cell is the only terminal state. The agent can move up, down, left, and right.

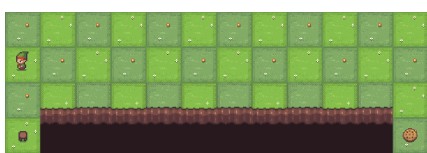

Figure 8: The Gymnasium CliffWalking environment.

## G.7 Bridge

The non-stationary bridge environment (Figure 9) is a grid-world setting where the agent must navigate from the starting cell to one of two goal cells. The environment was originally introduced by Lecarpentier and Rachelson [2019]. To reach a goal cell, the agent must cross a "bridge" surrounded by terminal cells. The secondary goal cell is farther from the starting location but less risky because fewer holes surround it. Unlike the CliffWalking environment, which has a single global transition probability, the left and right halves of the Bridge map each have separate probability distributions. NS-Gym allows for updates to just the left or right halves of the map or to the global value. Similar to the FrozenLake environment, if the agent moves in some direction, there is some probability that is moves in one of the perpendicular directions instead. The agent receives a +1 reward for

reaching a goal cell, a -1 reward for falling into a hole, and a 0 reward otherwise. Our version of the non-stationary bridge environment is not included in the standard Gymnasium Python package. We provide our implementation of the Bridge environment, as described by Lecarpentier and Rachelson [2019], as part of the NS-Gym package.

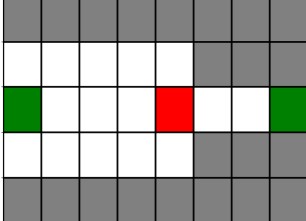

Figure 9: The Bridge environment. The start cell is in red, the two goals are in green, and the terminal "holes" are in gray.

## G.8 Ant

The Ant MuJoCo environment (Figure 10) is a 3D quadruped robot, where the agent must take actions to keep the ant in a healthy state while moving forward. NS-Gym can modify the gravity and mass of the ant's torso to induce non-stationary state transitions. The ant environment has a 105-dimensional continuous observation space and an eight-dimensional continuous action space.

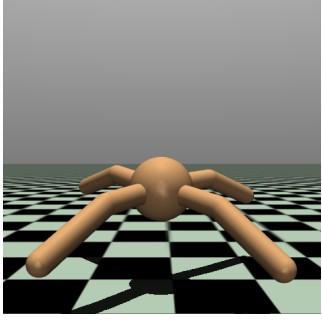

Figure 10: The Ant environment

## G.9 Half Cheetah

The Half Cheetah MuJoCo environment (Figure 11) NS-Gym can alter the environment's gravity, mass of the front and back shins, mass of the front and back thighs, mass of the front and back feet, and the damping coefficient of several leg joints. The half cheetah environment has a 17-dimensional continuous observation space and a six-dimensional continuous action space.

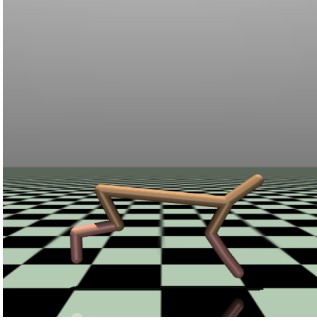

Figure 11: The Half Cheetah Environment

### G.10 Hopper

The Hopper MuJoCo environment is a two-dimensional one-legged figure. The agent's objective is to keep the hopper upright and moving forward. NS-Gym modifies the gravity, mass the the figure's torso, mass of its thigh, mass of its foot, the coefficient of friction of the floor, thigh joint damping coefficient, leg joint damping coefficient ,and the foot joint damping coefficient. The environment has an 11-dimensional continuous observation space and a three-dimensional action space.

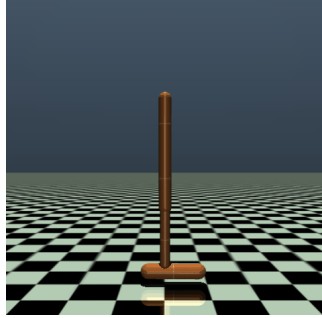

Figure 12: The Hopper Environment

### G.11 Inverted Pendulum

The Inverted Pendulum environment (Figure 13) is effectively the CartPole environment (G.1) powered by the MuJoCo simulation engine. The agent's objective is to keep the inverted pendulum upright. NS-Gym induces non-stationarity by altering gravity, the pole's mass, and the cart's mass. The environment has a four-dimensional continuous observation space and a one-dimensional continuous action space.

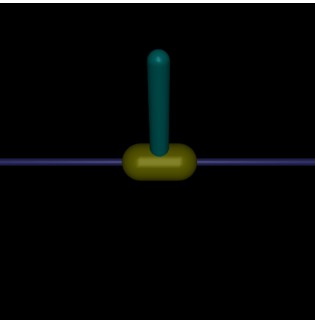

Figure 13: The Inverted Pendulum Environment

### G.12 Inverted Double Pendulum

The Inverted Double Pendulum environment (Figure 14) is a "double-pendulum" extension of the CartPole control task. The agent's objective is to keep the double pendulum upright. NS-Gym induces non-stationarity by modulating gravity, the cart mass, each double pendulum link mass, the damping coefficient of the joint connecting each pendulum link, and the damping coefficient of the joint connecting the pendulum to the cart. This environment has a nine-dimensional continuous observation space and a one-dimensional continuous action space.

### G.13 Reacher

The Reacher environment (Figure 15) is a two-jointed robotic arm where the agent has to maneuver the arm so that its end point is as close as possible to a target location. NS-Gym adds non-stationarity by altering each robotic arm link masses and the arm joint damping coefficients. This environment has a ten-dimensional continuous observation space and a two-dimensional continuous action space.

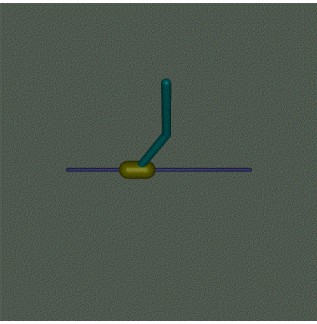

Figure 14: The Inverted Pendulum Environment

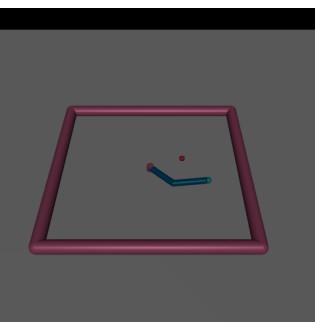

Figure 15: The Reacher Environment

### G.14  Swimmer

The Swimmer MuJoCo environment (Figure 16 is a "swimmer" that consists of three links. The agent needs to actuate the joints connecting each link to move the swimmer to the right as fast a possible. NS-Gym induces non-stationarity by altering the mass of the swimmer's middle segment. This environment has an eight-dimensional continuous observation space and a two-dimensional continuous action space.

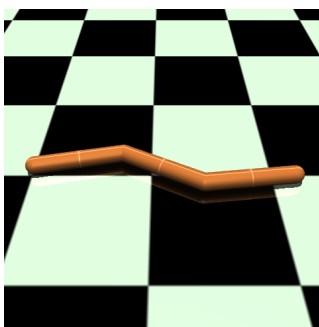

Figure 16: The Swimmer Environment

### G.15  Pusher

The Pusher MuJoCo environment (Figure 17) is a multi-jointed robotic arm. The agent needs to control the robot so that it moves a cylindrical object to a goal location. NS-Gym induces non-stationarity by modifying gravity, the mass of the shoulder link, the upper arm link mass, the forearm link mass, the shoulder joint damping coefficient, and the elbow flex joint damping coefficient. This environment has a 23-dimensional continuous observation space and a seven-dimensional continuous action space.

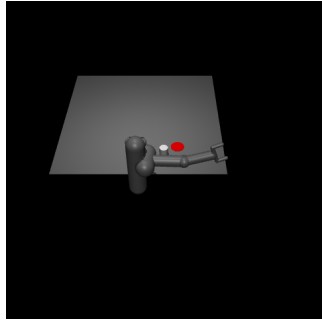

Figure 17: The Swimmer Environment

### G.16 Humanoid

The Humanoid MuJoCo environment (Figure 18) is a bipedal humanoid robot. The agent's goal is to actuate each of the robot joints to have it walk forward as fast as possible without falling. NS-Gym induces non-stationary dynamics by modulating gravity, torso mass, the pelvis mass, the masses of each part of the leg, the mass of the arms, the damping coefficient of each knee joint, and the damping coefficient of each elbow joint. This environment has a 348-dimensional continuous observation space and a 17-dimensional continuous action space. NS-Gym also supports the "stand-up" variation of the environment.

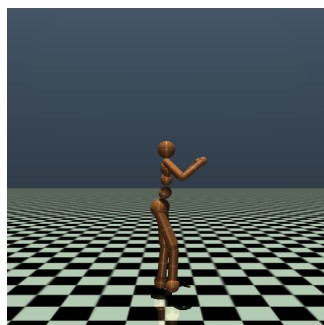

Figure 18: The Humanoid Environment

| Environment Name | Tunable Environmental Parameters |
|---|---|
| **AntEnv** | `gravity`, `torso_mass` |
| **HalfCheetahEnv** | `gravity`, `torso_mass`, `bthigh_mass`, `bshin_mass`, `bfoot_mass`, `fthigh_mass`, `fshin_mass`, `ffeet_mass`, `floor_friction`, `bthigh_damping`, `bshin_damping`, `bfoot_damping`, `fthigh_damping`, `fshin_damping`, `ffeet_damping` |
| **HopperEnv** | `gravity`, `torso_mass`, `thigh_mass`, `leg_mass`, `foot_mass`, `floor_friction`, `thigh_joint_damping`, `leg_joint_damping`, `foot_joint_damping` |
| **HumanoidEnv** | `gravity`, `torso_mass`, `lwaist_mass`, `pelvis_mass`, `right_thigh_mass`, `left_thigh_mass`, `right_shin_mass`, `left_shin_mass`, `right_foot_mass`, `left_foot_mass`, `right_upper_arm_mass`, `left_upper_arm_mass`, `right_lower_arm_mass`, `left_lower_arm_mass`, `right_knee_damping`, `left_knee_damping`, `right_elbow_damping`, `left_elbow_damping` |
| **InvertedPendulumEnv** | `gravity`, `pole_mass`, `cart_mass` |
| **InvertedDoublePendulumEnv** | `gravity`, `cart_mass`, `pole1_mass`, `pole2_mass`, `slider_damping`, `hinge1_damping`, `hinge2_damping` |
| **ReacherEnv** | `body0_mass`, `body1_mass`, `joint0_damping`, `joint1_damping` |
| **SwimmerEnv** | `body_mid_mass` |
| **PusherEnv** | `gravity`, `r_shoulder_pan_link_mass`, `r_shoulder_lift_link_mass`, `r_upper_arm_link_mass`, `r_forearm_link_mass`, `r_shoulder_pan_joint_damping`, `r_shoulder_lift_joint_damping`, `r_elbow_flex_joint_damping` |

Table 10: Environmental parameters as they appear in NS-Gym for MuJoCo environments.

## H Experimental Setup

In this section, we elaborate on how we set up the single and continuous change experiments for each environment.

### H.1 CartPole

- **Single update case**: We initialize the CartPole environment to its default state. After the first decision epoch, we increase the mass of the pole from $0.1$ to a value of $1.0$ and $1.5$.

- **Continuous update case**: We initialize the CartPole environment to its default state. After each decision epoch, we increase the mass of the pole by $0.1$.

We truncate the episode after $2500$ episode steps if the agent does not reach a terminal state.

### H.2 FrozenLake

- **Single update case**: We initially set the probability of moving in the intended direction to $0.7$ and the probability of moving in each perpendicular direction to $0.15$. After the first decision epoch, we change the probability of moving in the intended direction to $0.4$, $0.6$, or $0.8$. We update the chance of moving in a perpendicular direction accordingly.

- **Continuous update case**: We initialize the FrozenLake environment to be completely deterministic. We decrease the chance of moving in the intended direction by $0.2$ for the first three decision epochs. We update the chance of moving in a perpendicular direction accordingly.

We truncate the episode after $100$ episode steps if the agent does not reach a terminal state.

### H.3 CliffWalking

- **Single update case**: We initialize the environment to be deterministic. After the first decision epoch, we update the transition probability to a value of $0.8$, $0.6$, or $0.4$. The probability of moving in the perpendicular and reverse directions are updated accordingly.

- **Continuous update case**: We initialize the environment to be deterministic. For the first $10$ decision epochs, we decrease the chance of moving in the intended direction by $0.02$. The probabilities of moving in the perpendicular and reverse directions are updated accordingly.

In our experimental setup, we modify the standard CliffWalking rewards so that the goal state has a reward of +100. Additionally, after $200$ decision epochs, if the agent has not found the goal, we truncate the episode.

### H.4 Bridge

- **Single update case**: We initially set the probability of moving in the intended direction to $0.7$ and the probability of moving in each of the perpendicular directions to $0.15$. After the first decision epoch, we change the probability of moving in the intended direction to a value of $0.4$, $0.6$, or $0.8$. We update the chance of moving in a perpendicular direction accordingly.

- **Continuous update case**: We initialize the environment to be deterministic. At each decision epoch, the probability of going in the intended direction decreases by $0.1$.

We truncate the episode after $200$ steps if the agent does not reach a terminal state.

### H.5 Pendulum

- **Single update case**: We initialize the pendulum with a mass of $1.0$. After the first time step, we test two scenarios. In the first scenario, the pendulum's mass is increased to $1.5$. In the second scenario, the mass is increased to $2.0$.

- **Continuous update case**: The pendulum is initialized with a mass of $1.0$, and the mass is increased at each time step by $0.01$.

### H.6 Acrobot

- **Single update case**: The "first" link of the Acrobot double pendulum is initialized with a mass of $1.0$. After the first time step, we test two scenarios. In the first scenario, the link's mass is increased to $1.5$. In the second scenario, the mass is decreased to $0.5$.

- **Continuous update case**: The "first" link of the Acrobot is initialized with a mass of 1.0, and the mass is increased by 0.1 at each time step.

### H.7 MountainCar

- **Single update case**: The car's power is initialized to 0.0015. We then test two scenarios: one where the power supplied is halved and one where the power is doubled.

- **Continuous update case**: The car's power is updated continuously according to a geometric progression. Specifically, the power supplied at MDP time step $t$ is given by $0.0015 \cdot 0.9^t$.

## I  Algorithm Details

In this section, we provide descriptions of algorithms used in NS-MDP benchmarking experiments.

1) **MCTS** is an anytime online search algorithm that selects optimal action using a model of the environment. We use the Upper Confidence bound for Trees (UCT) algorithm [Kocsis and Szepesvári, 2006] with random rollouts.

2) The **AlphaZero** algorithm [Silver *et al.*, 2017] is a general game-playing algorithm that combines tree search with a deep value and policy neural network. We train the AlphaZero policy network on a stationary version and the environment but evaluate the agent on an NS-MDP.

3) We include the popular **DDQN** approach as a pure reinforcement learning method [van Hasselt *et al.*, 2015]. In the "with notification" experiments, we perform *some* gradient update steps using the most up-to-date model of the MDP (to resemble the baseline setting used by Pettet *et al.* [2024]).

4) **ADA-MCTS** as a heuristic tree search algorithm that learns the environmental dynamics and *acts as it learns* [Luo *et al.*, 2024]. ADA-MCTS uses a risk-averse strategy to explore the environment safely by balancing epistemic and aleatoric uncertainties. In our experiments, we only benchmarked ADA-MCTS when the updated environmental parameters are unavailable, as its core lies in learning about the updated change through environmental interactions.

5) The **RATS** algorithm proposed by Lecarpentier and Rachelson [2019] uses a minimax search strategy to act in a risk-averse manner to future environmental changes. The approach was originally designed against changes bounded by Lipschitz continuity.

6) We benchmark the **Policy-Augmented-MCTS** algorithm from Pettet *et al.* [2024], which computes a convex combination of returns generated through online search and a stale policy. Crucially, this combination occurs *outside the tree* (as opposed to the AlphaZero algorithm), thereby stabilizing search under non-stationarity. We evaluate PAMCTS with three $\alpha$ values, 0.25, 0.5, and 0.75, which control the extent to which the stale policy is preferred over online search.

The Tables 11, 12, 13, 14, 15, and 16 show the parameters used in each experiment. For PPO, DDPG, and A2C algorithms we reference the hyperparameters provided in the rl-baselines3-zoo Raffin [2020] repository.

**Single**

|     | Bridge | FrozenLake | CliffWalking | CartPole |
|-----|--------|------------|--------------|----------|
| $m$ | 500    | 300        | 1000         | 300      |
| $d$ | 100    | 100        | 200          | 500      |
| $c$ | $\sqrt{2}$ | $\sqrt{2}$ | $\sqrt{2}$ | $\sqrt{2}$ |
| $\gamma$ | 0.99 | 0.99   | 0.999        | 0.5      |

**Continuous**

|     | Bridge | FrozenLake | CliffWalking | CartPole |
|-----|--------|------------|--------------|----------|
| $m$ | 500    | 300        | 1000         | 300      |
| $d$ | 100    | 100        | 200          | 500      |
| $c$ | $\sqrt{2}$ | $\sqrt{2}$ | $\sqrt{2}$ | $\sqrt{2}$ |
| $\gamma$ | 0.99 | 0.99   | 0.999        | 0.5      |

Table 11: MCTS parameters for the single and continuous change experiments, where $m$ is the number of MCTS iterations, $d$ is the maximum rollout depth, $c$ is the exploration parameter, $\gamma$ is the tree discount factor.

|          | Bridge | FrozenLake | CliffWalking | CartPole |
|----------|--------|------------|--------------|----------|
| $m$      | 500    | 300        | 300          | 500      |
| $c$      | $\sqrt{2}$ | 1.44   | 1.44         | $\sqrt{2}$ |
| $\gamma$ | 0.99   | 0.999      | 0.999        | 1        |
| layers   | 3      | 3          | 3            | 2        |
| units    | 64     | 64         | 64           | 128      |
| $\alpha$ | 1      | 1          | 5            | 1        |
| $\epsilon$ | 0    | 0          | 0.75         | 0        |

Table 12: AlphaZero parameters for the single and continuous change experiments, where $m$ is the number of MCTS iterations, $c$ is the exploration parameter, $\gamma$ is the tree discount factor, layers are the number of hidden layers in the neural network, and units are the number of units in each hidden layer. The parameter $\alpha$ is the concentration parameter for the Dirichlet noise added to the priors in the root node of the search tree. The parameter $\epsilon$ controls the amount of noise added to the priors.

|        | Bridge | FrozenLake | CliffWalking | CartPole |
|--------|--------|-----------|-------------|----------|
| layers | 3      | 2         | 2           | 2        |
| units  | 64     | 64        | 128         | 64       |
| time   | 0.4    | 0.4       | 0.4         | 0.4      |

Table 13: DDQN parameters for both the single and continuous change experiments. The parameter layers are the number of hidden layers in the DDQN network. The parameter units are the number of units in each layer. In the "with" notification experiments, the time is the number of seconds the agent has to collect data and do gradient updates.

|            | Bridge | FrozenLake | CliffWalking | CartPole |
|------------|--------|-----------|-------------|----------|
| $m$        | 500    | 1000      | 1000        | 300      |
| $d$        | 200    | 500       | 200         | 500      |
| $c$        | $\sqrt{2}$ | $\sqrt{2}$ | $\sqrt{2}$ | $\sqrt{2}$ |
| $\gamma$   | 0.99   | 0.99      | 0.999       | 1        |
| layers     | 3      | 2         | 2           | 2        |
| units      | 64     | 64        | 128         | 64       |

Table 14: PAMCTS experiment parameters for single and continuous experiments, where $m$ is the number of MCTS iterations, $d$ is the MCTS search depth, $c$ is the exploration parameter, $\gamma$ is the discount factor, layers are the number of layers in the DDQN, and units are the number of units in each hidden layer.

|          | Bridge | FrozenLake | CliffWalking |
|----------|--------|-----------|-------------|
| $\gamma$ | 0.99   | 0.99      | 0.99        |
| $d$      | 3      | 3         | 3           |

Table 15: RATS algorithm parameters. $\gamma$ is the discount factor and $d$ is the tree search depth.

|          | Bridge | FrozenLake | CliffWalking |
|----------|--------|-----------|-------------|
| $\gamma$ | 0.99   | 0.99      | 0.99        |
| $m$      | 3000   | 100       | 3000        |

Table 16: ADA-MCTS algorithm parameters. $\gamma$ is the discount factor and $m$ is the number of iterations.

# J Experiment PAMCTS-Bound

For reference, we have computed the PAMCTS-Bound for the stochastic environments included in this paper. As a policy-agnostic metric it is one way to quantify the magnitude of non-stationarity itself, independent of any single agent's performance. This provides a measure of environmental difficulty other than reward alone.

| Environment | PAMCTS-Bound |
|---|---|
| FrozenLake - (Stochasticity $0.4$) | 0.3 |
| FrozenLake - (Stochasticity $0.6$) | 0.1 |
| FrozenLake - (Stochasticity $0.8$) | 0.1 |
| CliffWalking - (Stochasticity $0.4$) | 0.6 |
| CliffWalking - (Stochasticity $0.6$ | 0.4 |
| CliffWalking - (Stochasticity $0.8$ | 0.2 |

Table 17: PAMCTS-Bound for stochastic grid world experiments

# K  Experimental Results

In this section, we include additional experimental results and figures. Table 20 shows the complete results for the single change with and without notification experiments. Figures 19 , 20, 21, and 22 show the comparative performance of each decision-making agent in the single change experiments. Figures 23, 24, 25, and 26 show the comparative performance between all agents in the continuous change case.

We also provide additional experiments for the Pendulum, Acrobot, and Continuous MountainCar environments. All three environments have continuous action space, state space, or a sparse reward signal that makes it difficult for the tree-based approaches included in the NS-Gym baseline algorithms. We instead benchmark these environments with the RL methods using the Stable Baselines3 Raffin *et al.* [2021] implementations. Tables 18 and 19 present results for the PPO, DDPG and A2C algorithms in the single change and continuous change settings respectively.

Each algorithm was trained on the "default" settings for each environment. In the Pendulum environment, we modify the pendulum mass, in the Acrobot environment we modify the mass of one of the links and in the Mountain car environment, we modify the power applied to the car at each time step.

|  |  | PPO | DDPG | A2C |
|---|---|---|---|---|
| Pendulum | 1 | $-153.25 \pm 4.02$ | $-148.49 \pm 3.88$ | $-781.68 \pm 27.20$ |
|  | 1.5 | $-236.46 \pm 6.42$ | $-197.33 \pm 5.70$ | $-1248.56 \pm 9.61$ |
|  | 2 | $-321.14 \pm 8.86$ | $-677.28 \pm 8.73$ | $-1310.25 \pm 8.84$ |
| Acrobot | 1 | $-77.01 \pm 1.29$ | - | $-82.614 \pm 0.82$ |
|  | 1.5 | $-98.766 \pm 1.31$ | - | $-97.78 \pm 1.08$ |
|  | 0.5 | $-85.47 \pm 2.12$ | - | $-79.14 \pm 0.98$ |
| Mountain | 0.0015 | $-60.81 \pm 0.0015$ | $93.84 \pm 0.0063$ | $-9.18 \pm 0.0024$ |
| Car | 0.001125 | $-98.87 \pm 0.14$ | $88.74 \pm 0.14$ | $-99.9 \pm 0.00$ |
|  | 0.00225 | $-99.80 \pm 0.0068$ | $94.99 \pm 0.0048$ | $95.47 \pm 0.024$ |

Table 18: Additional experiments with single MDP change without notification on Pendulum, Acrobot, and continuous MountainCar environment where we have continuous state and action spaces. Blank spaces are where the policy is not applicable.

|  | PPO | DDPG | A2C |
|---|---|---|---|
| Pendulum | $-4095.16 \pm 5.74$ | $-5167.97 \pm 6.07$ | $-11333.77 \pm 53.04$ |
| Acrobot | $-187.12 \pm 2.47$ | - | $-227.442 \pm 1.68$ |
| MountainCar | $-98.89 \pm 0.11$ | $-90.69 \pm 0.31$ | $-99.98 \pm 0.00$ |

Table 19: Additional experiments where the MDP is continuously changing without notification on Pendulum, Acrobot and continuous MountainCar environment. These environments have either continuous action spaces or state spaces. Blank spaces are where the policy is not applicable

## Single Transition Change With and Without Notification

| | | MCTS | AlphaZero | DDQN | PAMCTS 0.25 | PAMCTS 0.5 | PAMCTS 0.75 | ADA-MCTS | RATS |
|---|---|---|---|---|---|---|---|---|---|
| **With Notification** | | | | | | | | | |
| Bridge | 0.4 | -0.28 ± 0.56 | -0.18 ± 0.1 | -0.82 ± 0.33 | -0.52 ± 0.29 | -0.12 ± 0.33 | -0.02 ± 0.33 | – | **0.34 ± 0.09** |
| | 0.6 | -0.32 ± 0.55 | **0.8 ± 0.06** | -0.80 ± 0.35 | -0.10 ± 0.33 | 0.3 ± 0.32 | 0.46 ± 0.3 | – | 0.30 ± 0.09 |
| | 0.8 | 0.32 ± 0.55 | **0.98 ± 0.02** | -0.90 ± 0.25 | 0.32 ± 0.2 | 0.84 ± 0.18 | 0.8 ± 0.2 | – | 0.08 ± 0.03 |
| FrozenLake | 0.4 | 0.09 ± 0.17 | 0.1 ± 0.03 | 0.2 ± 0.04 | 0.13±0.08 | 0.01 ± 0.02 | 0.07 ± 0.06 | – | 0.61 ± 0.05 |
| | 0.6 | 0.31 ± 0.27 | 0.21 ± 0.04 | 0.47 ± 0.05 | 0.34 ± 0.11 | 0.28 ± 0.11 | 0.35 ± 0.11 | – | **0.86 ± 0.04** |
| | 0.8 | 0.53 ± 0.29 | 0.51 ± 0.05 | 0.53 ± 0.05 | 0.62 ± 0.11 | 0.78 ± 0.10 | 0.66 ± 0.11 | – | **0.97 ± 0.02** |
| CliffWalking | 0.4 | -1767.75 ± 61.69 | **-588.23 ± 46.46** | -912.50 ± 42.39 | -1668.47 ± 64.08 | -1285.94 ± 71.43 | -1419.56 ± 68.83 | – | -1077.98 ± 48.82 |
| | 0.6 | -1162.91 ± 62.46 | **-0.48 ± 10.77** | -246.48 ± 2.08 | -1184.65 ± 57.88 | -495.81 ± 50.71 | -543.45 ± 54.80 | – | -400.72 ± 26.59 |
| | 0.8 | -846.64 ± 53.13 | **63.11 ± 3.53** | -20.89 ± 10.44 | -852.95 ± 56.15 | -43.06 ± 50.9 | -136.81 ± 25.46 | – | -245.54 ± 9.27 |
| CartPole | 1 | 633.62 ± 49.27 | 230.81 ± 1.06 | 92.8 ± 33.38 2 | 740.84 ± 43.23 | 122.89 ± 0.5 | 136.07 ± 0.29 | – | – |
| | 1.5 | 678.58 ± 51.13 | **902.05 ± 83.01** | 230.57 ± 21.39 | 702.58 ± 43.60 | 124.29 ± 0.47 | 135.22 ± 0.3 | – | – |
| **Without Notification** | | | | | | | | | |
| Bridge | 0.4 | -0.58 ± 0.47 | -0.26 ± 0.56 | -0.82 ± 0.33 | -0.58 ± 0.27 | -0.20 ± 0.33 | **-0.16 ± 0.33** | -0.54 ± 0.07 | -0.98 ± 0.02 |
| | 0.6 | -0.18 ± 0.57 | **0.58 ± 0.47** | -0.78 ± 0.36 | 0.46±0.33 | 0.46 ± 0.3 | 0.38 ± 0.31 | -0.16 ± 0.09 | 0.05 ± 0.08 |
| | 0.8 | 0.64 ± 0.45 | **0.92 ± 0.23** | -0.72 ± 0.4 | 0.4 ± 0.31 | 0.72 ± 0.23 | 0.8 ± 0.2 | 0.46 ± 0.09 | -0.01 ± 0.01 |
| FrozenLake | 0.4 | 0.11 ± 0.18 | 0.06 ± 0.02 | 0.22 ± 0.17 | 0.15 ± 0.04 | 0.16 ± 0.03 | 0.12 ± 0.03 | **0.67 ± 0.05** | 0.6 ± 0.05 |
| | 0.6 | 0.25 ± 0.25 | 0.25 ± 0.04 | 0.66 ± 0.19 | 0.3 ± 0.05 | 0.33 ± 0.05 | 0.27 ± 0.04 | 0.56± 0.05 | **0.88 ± 0.03** |
| | 0.8 | 0.53 ± 0.29 | 0.39 ± 0.05 | 0.91 ± 0.12 | 0.74 ± 0.04 | 0.68 ± 0.05 | 0.54 ± 0.05 | 0.49 ± 0.05 | **0.97 ± 0.02** |
| CliffWalking | 0.4 | -1593.89 ± 68.9 | -543.94 ± 45.98 | -1742.54 ± 91.29 | -1572.21 ± 60.82 | **-477.50 ± 54.66** | -1382.04 ± 77.88 | -1503.34 ± 53.57 | -777.55 +/- 31.19 |
| | 0.6 | -1216.72 ± 63.68 | **6.97 ± 8.2** | -1018.27 ± 96.95 | -1159.77 ± 53.85 | -374.64 ± 44.31 | -477.50 ± 54.65 | -1019.72 ± 35.99 | -314.84 ± 12.8 |
| | 0.8 | -773.62 ± 54.67 | **64.41 ± 3.44** | -287.17 ± 40.55 | -790.60 ± 46.66 | -54.22 ± 14.25 | -109.08 ± 25.99 | -523.73 ± 23.79 | -231.86 ± 4.22 |
| CartPole | 1 | **600.90 ± 47.68** | 441.1 ± 51.96 | 135.53 ± 0.28 | 525.98 ± 31.91 | 120.48 ± 0.57 | 135.41 ± 0.32 | – | – |
| | 1.5 | **641.28 ± 50.47** | 272.82 ± 21.25 | 139.19 ± 0.27 | 467.35 ± 25.11 | 117.60 ± 1.24 | 135.42 ± 0.34 | – | – |

Table 20: Table of mean rewards and standard error across for the single change environmental parameter change experiment. The best-performing agents for each environment are in bold.

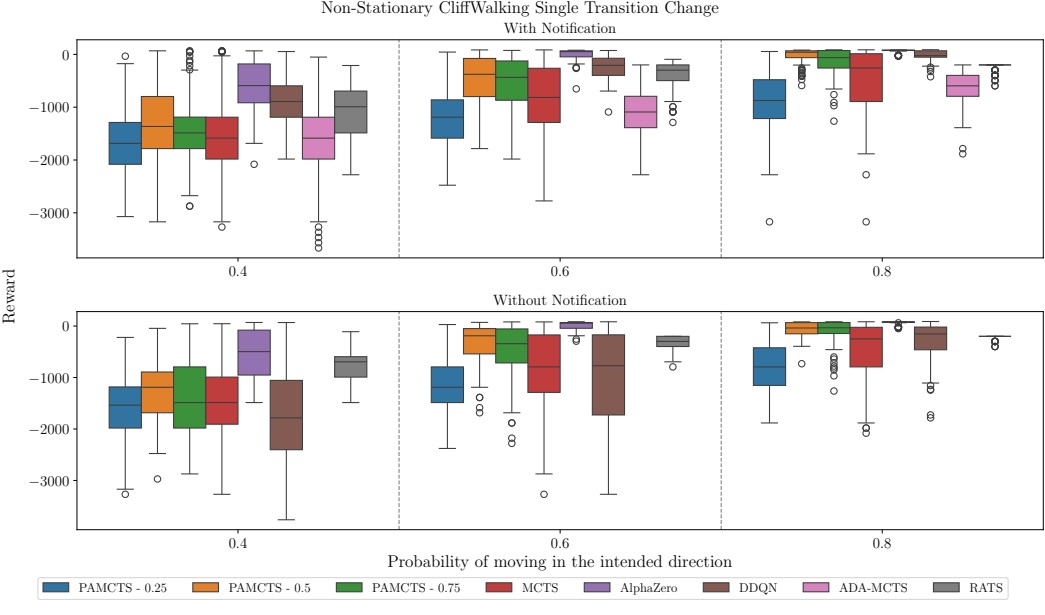

Figure 19: Distribution of rewards for the CliffWalking experiments with a single change.

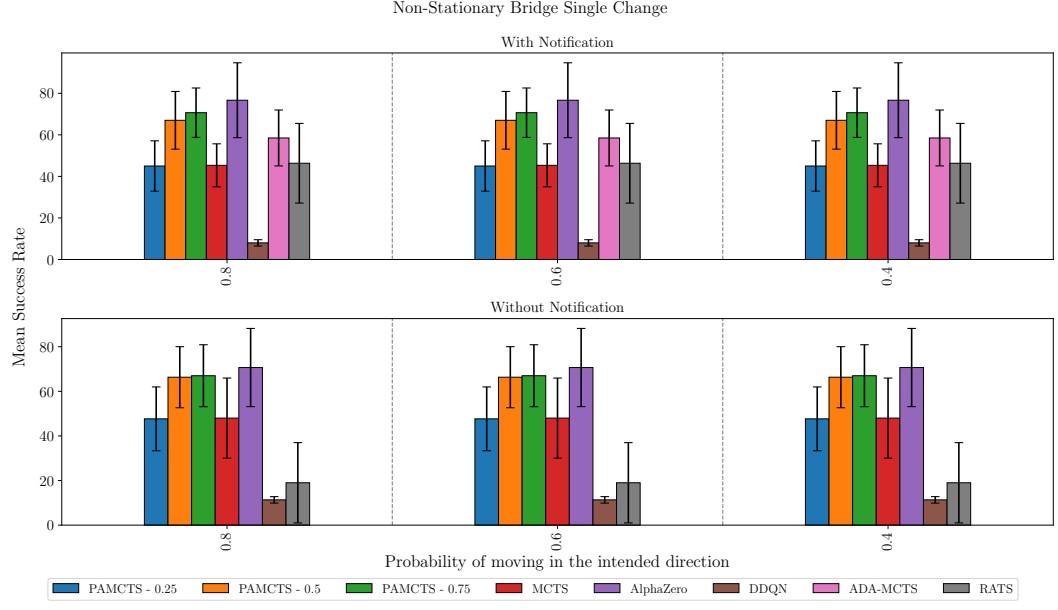

Figure 20: Average success rate (i.e., the agent finds the goal state) for each agent in the single change experiments.

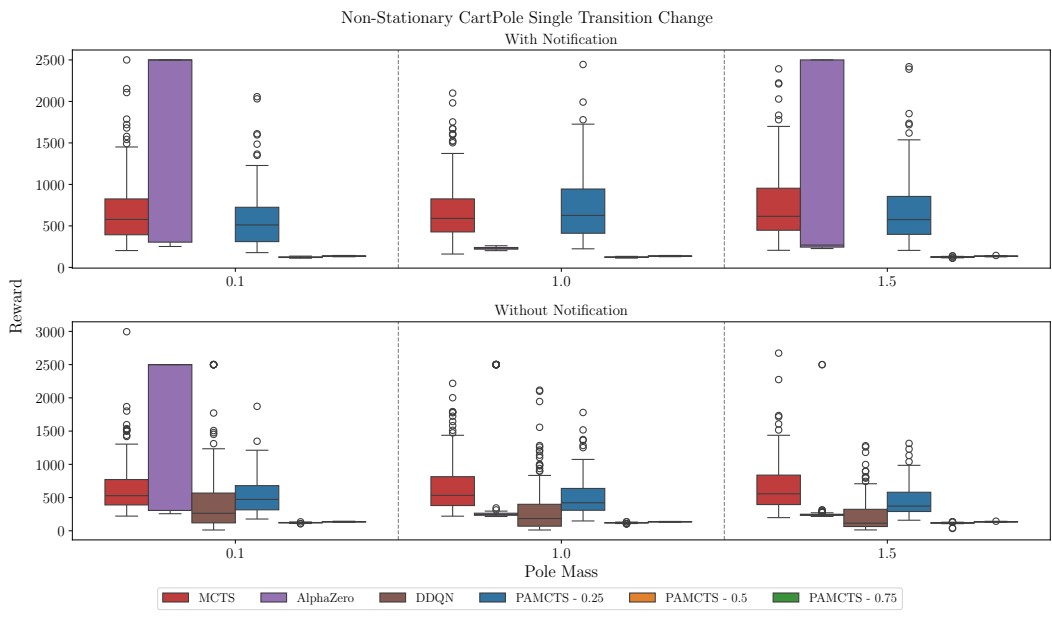

Figure 21: Distribution of episode rewards for each agent tested on non-stationary CartPole environment with and without notification.

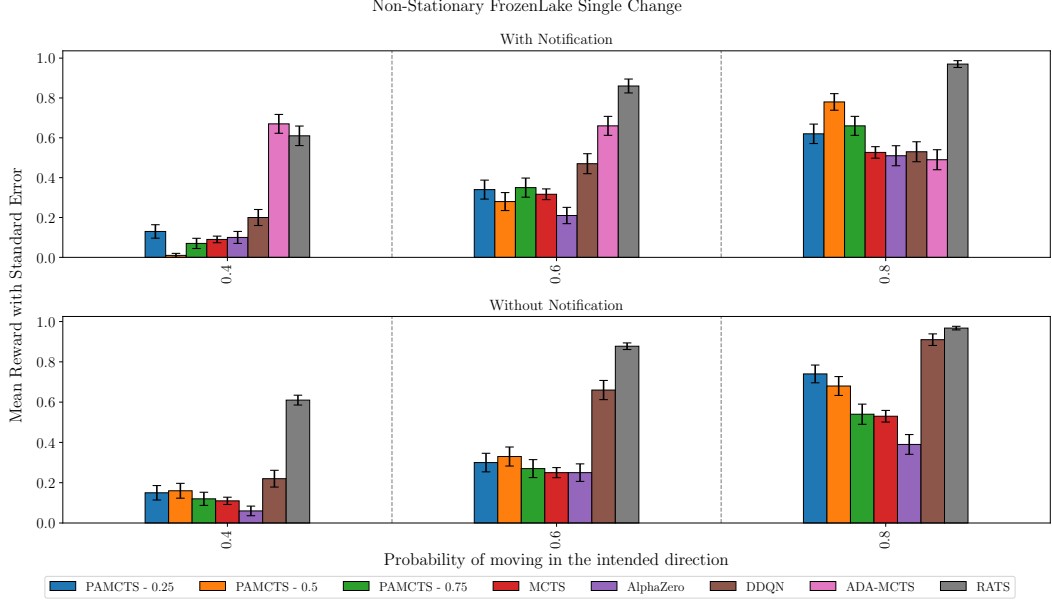

Figure 22: Mean episode reward and standard error for each agent in a non-stationary FrozenLake environment with a single change in its transition function.

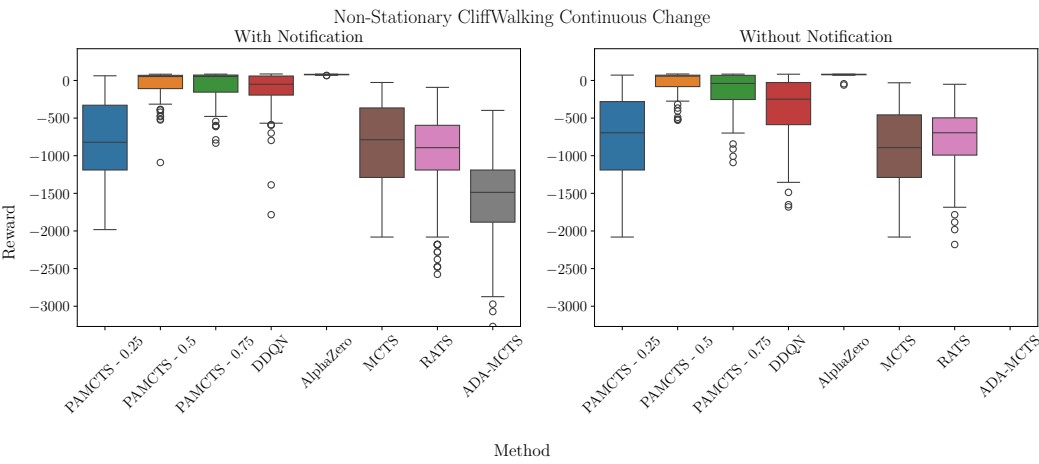

Figure 23: Distribution of episode reward for each agent under the continuous change experiment conditions.

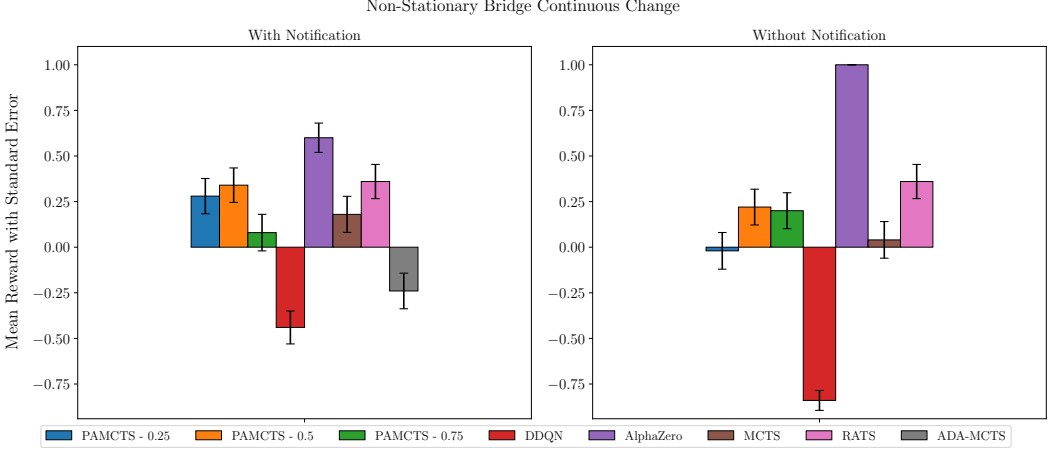

Figure 24: Mean reward and standard error for agents in the non-stationary Bridge environment under the continuous change conditions.

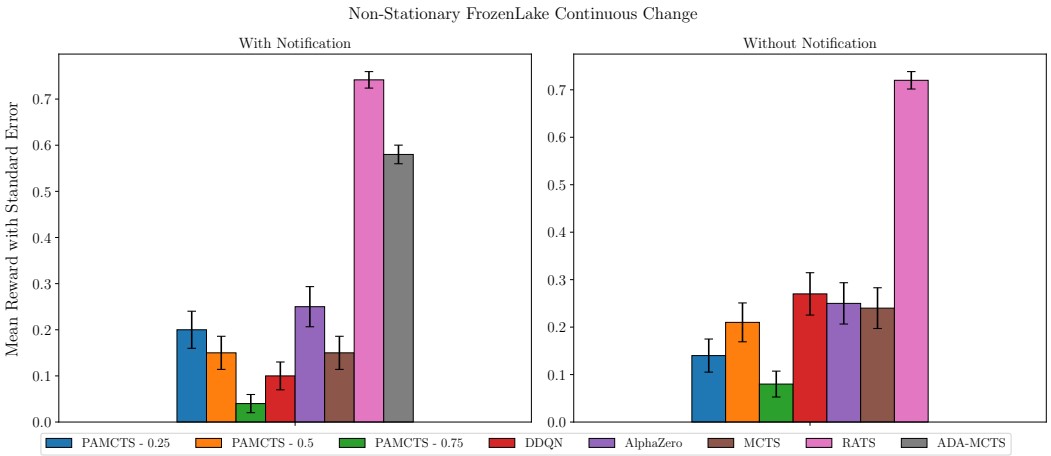

Figure 25: Mean reward and standard error for agents in the non-stationary FrozenLake environment under continuous change conditions.

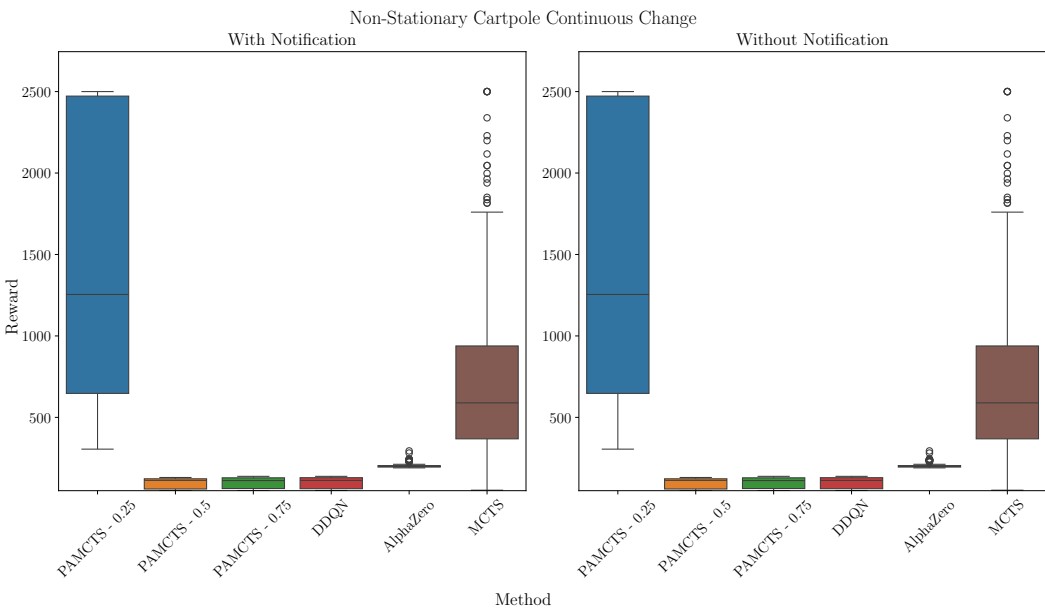

Figure 26: Distribution of episode rewards for agents in the continuous non-stationary CartPole environment.

