# OpenReview forum: "NS-Gym: A Comprehensive and Open-Source Simulation Framework for Non-Stationary Markov Decision Processes"
_NeurIPS.cc/2025/Datasets_and_Benchmarks_Track — NeurIPS 2025 Datasets and Benchmarks Track poster_

### Official Review · Reviewer_xzw8 · 2025-06-22

**Rating:** 4
**Confidence:** 4

**Summary:**

This paper proposes NS-Gym, a benchmark aims at evaluating deep neural network control policies learned by Reinforcement Learning (PPO, DDQN, and others) in the NS-MDP settings, in which the parameters of the transit dynamics are time-variant. The benchmark includes several classical control tasks (e.g. CartPole, Bridge, CliffWalking, FronzenLake, Pendulum). The motivation to provide a structured and accessible platform for evaluating the deep neural network control policies at NS-MDPs is appreciated and the environment design reflects some of the timing and architectural limits to the current RL-based policy learning methods since they do not originally consider the time invariant parameters. However, the paper raises multiple critical concerns regarding novelty, depth, and justification.

**Dataset Code Accessibility:**

Yes

**Ethical Considerations:**

No, there are no or only very minor ethics concerns

**Final Justification:**

I have carefully reviewed the authors' responses to the other reviewers, particularly Reviewer xv2q. I find that the authors have addressed several of the concerns I raised, especially regarding experimental rigor and the complexity of the benchmark tasks.

Although their responses were not directed to me, they do clarify key points relevant to my original concerns. In recognition of this, I would like to adjust my score accordingly.

**Limitations Weaknesses:**

1. Lack of Clear Novelty: While the paper claims to introduce a “comprehensive benchmark,” it primarily adapts standard Gym-style tasks to support deep reinforcement learning-based agents. The core environments, task structures, and reward functions seem to be extended from existing work by making the parameters in time-variant manner. Maybe this is a difficult point. Please explain the challenging issue of the extensions in detail.

2. Simplistic Models and Limited Challenge: The tasks in NS-Gym are relatively simple (e.g., 2D stabilization and tracking) and do not reflect the complexity or dimensionality often encountered in real applications. It will be appreciated that the authors can at least remark how the tasks will be different when dimensionality goes high, or the low dimensionality cases are enough to show sufficient evaluation results. Or maybe this work is still in an initial stage, and high-dimensional cases will be the future work?

3. Extensibility and Usability Not Demonstrated: Although the benchmark supports a few deep RL algorithms, such as PPO, the paper does not discuss how one might integrate new architectures or learning methods beyond what is provided. There is no explanation of modularity, task definition APIs, or compatibility with mainstream RL toolkits. This raises questions about whether NS-Gym is a general-purpose benchmark or simply a demo suite for selected methods.

4. Fidelity Claims Are Unsubstantiated: The paper suggests that NS-Gym environments reflect deep RL control requirements, but there is no empirical validation or comparative analysis. For example, why do different algorithms perform differently in different types of time-variant settings? Besides, some methods are not developed for NS-MDPs , and it is important to highlight the reasons for the failures and successes in the NS-MDPs.

5. Non-Stationary Setting: It will be good if the authors can summarize some insights for RL algorithms design from the experience of establishing NS-Gym. For example, how do RL algorithms for Stationary MDPs perform in the case that the parameters change smoothly on a time scale? How about the case where the change is not smooth?

**Strengths Contributions:**

(1) The paper addresses an underexplored area: benchmarks for evaluating the performance of learned policies (especially neural network control policies) under NS-MDP settings.

(2) It provides a working implementation with several environments and compatible baselines.

(3) The domain motivation is timely and relevant to the deep reinforcement learning community.

---

> ### Author Rebuttal · Authors · 2025-07-31
>
> We have flagged to the AC that this review has nothing to do with our paper, as it repeatedly points out "evaluating neuromorphic control policies, particularly those based on spiking neural networks (SNNs)." Our paper has no connection to neuromorphic control or spiking neural networks (SNNs); in fact, these phrases are not mentioned in our paper at all. Our paper is on non-stationary Markov decision processes.
>
> It is possible that this review was for a different paper assigned to the reviewer.

---

> > ### Comment · Reviewer_xzw8 · 2025-08-01
> > **Statement on My Previous Review and Apologies**
> >
> > Dear Authors,
> >
> > I have updated my review comments by sending an official comment.
> >
> > I sincerely apologize for the confusion and inconvenience my earlier comments may have caused to you. In preparing the previous version, I used translation software to convert my review from my native language into English and failed to check the output carefully. This was my mistake, and I genuinely regret the oversight.
> >
> > I am truly sorry. I understand that the limited time remaining may not allow you to fully address the revised concerns. I will carefully consider your response to my updated comments and take the time constraints into account when deciding whether to adjust my score during the discussion period.
> >
> > Best regards,

---

> > > ### Comment · Area_Chair_Y3Kz · 2025-08-02
> > > **Problematic Review**
> > >
> > > Thank you for clarifying your review.
> > >
> > > We appreciate that reviewing for NeurIPS is voluntary and we appreciate your efforts to make clarification.  This helps us better understand what happened with the initial review and will help us as we go forward in the development of our policies around the use of LLMs in review writing.
> > >
> > > In our current authorship policies, we state:
> > >
> > > 3. Do I need to declare LLM usage if it’s just for writing or formatting?
> > > No, if the LLM is used only for writing, editing, or formatting purposes and does not impact the core methodology, scientific rigorousness, or originality of the research, declaration is not required.
> > >
> > > Unfortunately, your use of an LLM in writing your review introduced material changes which we believe impacted the scientific rigorousness of the original review.  While we hope that the authors can benefit from your revised review, in all fairness, we cannot hold them to the same requirement to respond.
> > >
> > > Please do not expect a rebuttal from the authors.  We need to down-weight this review given the initial confusion.  We do note that several of the concerns you raised are echoed in the other reviewers comments and hopefully the authors have addressed them.  As per the spirit of Open Review, we welcome any information that can help improve papers but we have decided that it would be unfair to hold the authors to the same obligation to respond in a shortened time frame.
> > >
> > > Best Regards,
> > >
> > > AC

---

> > > > ### Comment · Reviewer_xzw8 · 2025-08-03
> > > > **Further Clarification**
> > > >
> > > > Dear Area Chair,
> > > >
> > > > Thank you for your explanation.
> > > >
> > > > I fully understand and accept the decision to down-weight my review, given the initial confusion it may have caused. It would also be unreasonable to expect the authors to respond within a shortened timeframe due to my revision.
> > > >
> > > > I have carefully reviewed the authors' responses to the other reviewers, particularly Reviewer xv2q. I find that the authors have addressed several of the concerns I raised, especially regarding experimental rigor and the complexity of the benchmark tasks.
> > > >
> > > > Although their responses were not directed to me, they do clarify key points relevant to my original concerns. In recognition of this, I would like to adjust my score accordingly. I understand that this adjustment may also be down-weighted, but I hope it still provides some context for your evaluation.
> > > >
> > > > Sincerely,

---

> > ### Comment · Reviewer_xzw8 · 2025-08-03
> > **Acknowledgement and Apology**
> >
> > Dear Authors,
> >
> > Thank you for your thoughtful responses in the rebuttal to other reviewers.
> > Since my revised comments were submitted late, I totally understand that it is not reasonable to ask the authors to reply to my revised comments.
> >
> > I have carefully read your replies to the other reviewers, particularly your clarifications to Reviewer xv2q.
> > Your explanations regarding the experimental design and the complexity of the benchmark tasks helped address several of the concerns I initially raised.
> > While the responses were not directed to me specifically, they nonetheless clarified key aspects relevant to my review.
> >
> > In light of this, I will adjust my score to better reflect my updated assessment. I also want to reiterate my sincere apologies for the confusion caused by the initial review.
> >
> > Sincerely,

---

> > > ### Comment · Area_Chair_Y3Kz · 2025-08-03
> > > **Thank you for understanding**
> > >
> > > Dear Reviewer,
> > >
> > > Thank you for understanding.  I wish I had caught the error earlier and we had been able to resolve it before reviews went out to authors, but unfortunately I missed it.
> > >
> > > Best Regards,
> > >
> > > AC

---

> > > > ### Comment · Reviewer_xzw8 · 2025-08-03
> > > >
> > > > Dear Area Chair
> > > >
> > > > I understand that you have been very busy.
> > > > This is my responsibility.
> > > > I would like to do everything I could to support.
> > > >
> > > > Best,

---

> ### Comment · Reviewer_xzw8 · 2025-08-01
> **Updated Review Comments**
>
> **Summary:**
>
> This paper proposes NS-Gym, a benchmark aims at evaluating deep neural network control policies learned by Reinforcement Learning (PPO, DDQN, and others) to the NS-MDP settings, in which the parameters of the transit dynamics are time-variant. The benchmark includes several classical control tasks (e.g. CartPole, Bridge, CliffWalking, FronzenLake, Pendulum). The motivation to provide a structured and accessible platform for evaluating the deep neural network control policies at NS-MDPs is appreciated and the environment design reflects some of the timing and architectural limits to the current RL-based policy learning methods since they do not originally consider the time invariant parameters. However, the paper raises multiple critical concerns regarding novelty, depth, and justification.
>
> **Strengths Contributions:**
>
> (1) The paper addresses an underexplored area: benchmarks for evaluating the performance of learned policies (especially neural network control policies) under NS-MDP settings.
>
> (2) It provides a working implementation with several environments and compatible baselines.
>
> (3) The domain motivation is timely and relevant to the deep reinforcement learning community.
>
> **Limitations Weaknesses:**
>
> 1. Lack of Clear Novelty: While the paper claims to introduce a “comprehensive benchmark,” it primarily adapts standard Gym-style tasks to support deep reinforcement learning-based agents. The core environments, task structures, and reward functions seem to be extended from existing work by making the parameters in time-variant manner. Maybe this is a difficult point. Please explain the challenging issue of the extensions in detail.
>
> 2. Simplistic Models and Limited Challenge: The tasks in NS-Gym are relatively simple (e.g., 2D stabilization and tracking) and do not reflect the complexity or dimensionality often encountered in real applications. It will be appreciated that the authors can at least remark how the tasks will be different when dimensionality goes high, or the low dimensionality cases are enough to show sufficient evaluation results. Or maybe this work is still in an initial stage, and high-dimensional cases will be the future work?
>
> 3. Extensibility and Usability Not Demonstrated: Although the benchmark supports a few deep RL algorithms, such as PPO, the paper does not discuss how one might integrate new architectures or learning methods beyond what is provided. There is no explanation of modularity, task definition APIs, or compatibility with mainstream RL toolkits. This raises questions about whether NS-Gym is a general-purpose benchmark or simply a demo suite for selected methods.
>
> 4. Fidelity Claims Are Unsubstantiated: The paper suggests that NS-Gym environments reflect deep RL control requirements, but there is no empirical validation or comparative analysis. For example, why do different algorithms perform differently in different types of time-variant settings? Besides, some methods are not developed for NS-MDPs , and it is important to highlight the reasons for the failures and successes in the NS-MDPs.
>
> 5. Non-Stationary Setting: It will be good if the authors can summarize some insights for RL algorithms design from the experience of establishing NS-Gym. For example, how do RL algorithms for Stationary MDPs perform in the case that the parameters change smoothly on a time scale? How about the case where the change is not smooth?

---

### Official Review · Reviewer_xv2q · 2025-06-23

**Rating:** 5
**Confidence:** 2

**Summary:**

This paper presents NS-Gym, an open-source simulation toolkit for Non-Stationary Markov Decision Processes (NS-MDPs), built as a wrapper around the Gymnasium library.

**Additional Feedback:**

- Q1: A benchmark paper must provide insights. I strongly suggest deepening your analysis to explain why different agents succeed or fail under specific types of non-stationarity.
- Q2: The utility of a new tool must be clearly demonstrated. Future work should test the framework on more complex, high-dimensional tasks to prove its scalability and relevance.
- Q3: I recommend using the evaluation metrics that the authors designed in Section 3.2 to support the analysis, as this would add much-needed depth.

**Dataset Code Accessibility:**

Yes

**Dataset Code Comments:**

The code for the framework and experiments is provided in the supplementary material with setup and details.

**Ethical Considerations:**

No, there are no or only very minor ethics concerns

**Final Justification:**

The authors have comprehensively addressed the critical concerns raised in the review, significantly strengthening the paper’s contribution and scientific impact. Key resolved issues include:
﻿
1. Integration of the "Ant" environment (a high-dimensional control task) into NS-Gym, showcasing the framework’s ability to handle complex, realistic non-stationary scenarios.
2. Introduction of policy-agnostic PA-MCTS bounds (based on [2401.03197]) to quantify non-stationarity, providing a novel metric for environment difficulty and enabling deeper analysis beyond reward-based comparisons.
3. Preliminary results on the Ant environment (e.g., gravity modulation with PPO) demonstrate the framework’s utility in studying non-stationary dynamics, aligning with the goal of generating actionable insights for algorithm design.
﻿
These revisions transform the work from a tool description into a robust benchmark toolkit with clear scientific value, justifying the higher score. The authors’ proactive additions directly resolve prior limitations.

**Limitations Weaknesses:**

1. The primary weakness is the shallow experimental analysis. For a benchmark paper, simply presenting performance tables is not enough. The expectation is that the benchmark will be used to generate novel scientific insights.
2. The experiments are confined to simple, low-dimensional domains such as CartPole and FrozenLake. This limited scope makes it difficult to assess the framework's utility and scalability for more complex and relevant problems in modern reinforcement learning.

**Strengths Contributions:**

1. The work provides a tool that addresses the recognized need for better benchmarking standards in non-stationary reinforcement learning.
2. Building the framework upon Gymnasium is a practical choice that promotes accessibility.

---

> ### Author Rebuttal · Authors · 2025-07-31
>
> Thank you for your time and valuable feedback. We address your concerns below.
>
> While we agree that demonstrating the framework's utility on high-dimensional problems is important, one of our core goals was to create a standardized, reproducible, and modular experimental toolkit for non-stationary MDPs. Note that this fundamental challenge is distinct from the dimensionality of the base environment, and clearly validated in well-understood environments like CartPole and FrozenLake.
>
> Based on the reviewer’s feedback, we are integrating the Mujoco suite of environments into NS-Gym, which are higher-dimensional control problems with complex dynamics. During the rebuttal period, we have already integrated the “Ant” environment, which has a 108-dimensional continuous observation space. In this scenario, we use a `ContinuousScheduler` with the `OscillatingUpdate` function to smoothly modulate the environment's gravity according to a sinusoidal wave. We trained a PPO agent on the default stationary dynamics and evaluated it on this new non-stationary instance without notifications.
>
> | Model | Stationary Env Reward | Non-Stationary Env Reward |
> | --- | --- | ---|
> | PPO | $778.13 \pm 69.1$ | $416.11 \pm 39.51$|
>
> We will add more Mujoco environments in the final version of the paper. As we continue to maintain the package, we will expand this benchmark to new standardized non-stationary problem domains.
>
> For reference, we have computed the PA-MCTS (Pettet et. al, 2024) bound for the stochastic environments included in the paper. As a policy-agnostic metric, it is one way to quantify the magnitude of the non-stationarity itself, independent of any single agent’s performance.  This provides another measure of environment difficulty than reward alone.
>
> |Environment	|PA-MCTS Bound |
> |--- |---|
> FrozenLake – (Stochasticity 0.4)|	0.3
> FrozenLake – (Stochasticity 0.6)|	0.1
> FrozenLake – (Stochasticity 0.8)|	0.1
> Cliffwalking – (Stochasticity  0.4)|	0.6
> Cliffwalking – (Stochasticity 0.6)	|0.4
> Cliffwalking – (Stochasticity 0.8)	|0.2
>
> **Refrences**
>
> Pettet, A., Zhang, Y., Luo, B., Wray, K., Baier, H., Laszka, A., Dubey, A., & Mukhopadhyay, A. (2024). Decision making in non-stationary environments with policy-augmented search. arXiv. https://arxiv.org/abs/2401.03197

---

> > ### Comment · Reviewer_xv2q · 2025-08-02
> >
> > The authors have made significant improvements to address the key concerns raised in the review. By integrating the "Ant" environment (a high-dimensional control task with 108-dimensional observations) into NS-Gym and demonstrating non-stationary dynamics via gravity modulation, they directly tackle the scalability limitation highlighted in the original critique. Additionally, the introduction of PA-MCTS bounds as a policy-agnostic metric for quantifying non-stationarity provides a novel analytical tool, aligning with the request for deeper scientific insights. While further experiments on more complex environments are planned, the current additions substantiate the framework's utility and theoretical grounding, addressing the core weaknesses of limited evaluation and analysis.

---

### Official Review · Reviewer_xsub · 2025-07-03

**Rating:** 4
**Confidence:** 4

**Summary:**

This paper introduces a non-stationary layer on top of the OpenAI Gym environment to simulate non-stationary Markov Decision Process (MDP) problems. The resulting framework is both general and flexible. Using this framework, the authors evaluate eight environments (Bridge, FrozenLake, CliffWalking, CartPole, Pendulum, ContinuousMountainCar, MountainCar, and Acrobot) under various reinforcement learning algorithms, including MCTS, DDQN, AlphaZero, ADA-MCTS, RATS, and PAMCTS, all in non-stationary settings. The experiments demonstrate the framework’s usefulness. Additionally, the tutorial is clearly written.

**Dataset Code Accessibility:**

Yes

**Dataset Code Comments:**

The tutorial is well written, and the repo is well organized.

**Ethical Considerations:**

No, there are no or only very minor ethics concerns

**Final Justification:**

The authors have addressed my concerns and I kept my score unchanged.

**Limitations Weaknesses:**

1. **Computational Overhead:** The computational times appear to be a concern, as shown in Appendix D. For example, the CliffWalking environment experiences an overhead of approximately 800x. Could the authors elaborate on the reasons behind this significant overhead?

2. **Parallelization and Vectorization:** Could the authors provide more details on the associated overhead in the case of parallelization and vectorization?

3. **Mathematical Description of \$\theta\_t\$:** The evolution of \$\theta\_t\$ is described somewhat vaguely in the current manuscript. The authors may want to provide a clear mathematical formulation of how \$\theta\_t\$ evolves over time.

4. **Simulating Nonstationary MDPs for Experimentation and Offline Policy Evaluation:** Simulating nonstationary MDPs is also an important topic for experimentation and offline policy evaluation. Could the authors comment on how such problems could be incorporated into the proposed framework (e.g., \[1, 2])?

   * \[1] Hu, Yuchen, and Stefan Wager. "Switchback experiments under geometric mixing." *arXiv preprint* arXiv:2209.00197 (2022).
   * \[2] Johari, R., Peng, T., & Xing, W. (2025). Estimation of Treatment Effects Under Nonstationarity via Truncated Difference-in-Q's. *arXiv preprint* arXiv:2506.05308.

**Strengths Contributions:**

1. Non-stationary MDP is an important trend in developing RL algorithms that work in the real world. So this NS-GYM package can be potentially impactful.

2. The benchmark experiments are thorough, and some interesting insights about notification/without notification are revealed.

---

> ### Author Rebuttal · Authors · 2025-07-31
>
> Thank you for the insightful comments and questions, and for appreciating the potential impact of NS-Gym. We address each of your points below.
>
> ## Computational Overhead:
>
> We acknowledge the computational inefficiency in the CliffWalking implementation of the submitted version. In the submitted version of the package, the transition function was recomputed after every parameter update and stored in a table. This oversight occurred before running the baseline experiments, and for the sake of consistency, we opted not to refactor before the submission deadline.
>
> Based on the reviewer’s comment, **we have optimized the environment, making step execution time 67x faster.** Please refer to the table below for updated step times, with parameter updates applied at every time step. We achieved this with numpy vectorization tricks and updating transition probabilities in place.
>
> | Environment | Gymnasium ($10^{-6}s$) | NS-Gym ($10^{-6}s$) | Absolute Overhead ($10^{-6}s$) |
> | --- | --- | --- | --- |
> | ClffWalking | $10.13 \pm 0.01$ | $124.50 \pm 0.068$ | $114.37 \pm 0.07$  |
>
> ## Parallelization and Vectorization:
>
> Please see the asynchronous vectorized execution experiment benchmarks below.  For each environment, we ran 8 parallel vectorized environments for a total of 800,000 steps across all environments.  We report environment steps per second (SPS).
>
> |Environment |Gym (SPS) |NS-Gym (SPS) |
> | --- |--- | ---|
> Cartpole	| $86,806.16 \pm 1,025.21$	|$42,813.68 \pm 2,011.52$|
> Pendulum	|$54,259.26 \pm 216.82$ |$31,677.50 \pm 1,493.39$|
> Acrobot	|$39,308.93 \pm 1,214.93$	|$29,803.10 \pm 965.21$|
> Mountaincar	|$82,620.94 \pm 898.42$	|$42,802.29 \pm 2,026.04$|
> FrozenLake	|$83,421.50 \pm 770.88$	|$17,768.18 \pm 428.61$|
> CliffWalking	|$83,978.50 \pm 626.08$	|$22,917.84 \pm 567.25 $|
>
> ## Mathematical Description of Theta:
>
> $\theta$ denotes the set of environmental variables that affect the transition function. We use this notation, as in the seminal work by Campo et al. (1991), to clearly segregate the evolution of the environmental parameters that characterize non-stationarity from the agent’s decision-making module. However, by design, the evolution of $\theta$ is left to the end user; it is completely configurable for a diverse set of non-stationary environments. At a high level, NS-Gym defines
> 1) A transition function $P(s’| s,a,\theta)$ that defines the probability of transitioning to state $s’$ given the current state, action taken, and set of environmental parameters $\theta$.
> 2) A function $f(t)$ i.e $\theta_{t+1} = f(\theta_t,t)$  that controls how $\theta$ evolves over time.
>
> In our framework, we implement $f(t)$ as the “schedulers,” which determine when updates occur, and “parameter update functions,” which determine how $\theta$ changes. This design allows for flexible control over the nature of the non-stationarity.
>
> ## Simulating Nonstationary MDPs for Experimentation and Offline Policy Evaluation:
>
> While NS-MDPs primarily capture the problem of online adaptation (as opposed to contextual MDPs or POMDPs), NS-Gym is flexible enough to replicate the setups in the papers cited by the reviewer (Hu & Wager, 2022; Johari et al., 2025). The general approach involves wrapping the core simulation logic of the problem domain within a standard Gymnasium environment and using NS-Gym to manage the non-stationarity.
>
> Here is a quick walkthrough on how we can use NS-Gym to model the specific experimental setups you mentioned.
>
> **Switchback Experiment in Hu and Wagner (2022)**: The base NS-MDP in this experiment is parameterized by a hidden state H and a market condition M. NS-Gym can model these distinct components effectively:
>
> - Market Condition (M): This showcases NS-Gym's ability to separate when a parameter changes from how it changes. A built-in `RandomScheduler` can be configured to trigger a market condition switch with a probability of 0.5 at each timestep. In response, a simple custom `UpdateFunction` would draw a new value for M from the set {1,2,3}.
> - Hidden State (H): Since the evolution of H is a function of the agent's intervention W (passed as an action), its transition logic would be implemented directly within the environment's `step()` method. This is the standard approach for state dynamics that depend on both environmental conditions (M) and agent actions (W).
>
> **Ride-Sharing and Hospital Simulators (Hu & Wager, 2022; Johari et al., 2025)**:
> - The core simulator logic (e.g., driver/rider generation from the NYC taxi dataset, or patient arrival and queuing logic) must first be encapsulated within a custom Gymnasium environment.
> - NS-Gym can then be used to control the non-stationarity. For instance, it can dynamically regulate driver shift times and request rates in the ride-sharing scenario or update the patient arrival-rate parameter in the hospital simulation based on real-world data traces.
>
> We thank you again for your time and valuable feedback. We will add these examples to the tutorial in the final version of the paper.
>
> **References**
>
> Campo, L., Mookerjee, P., & Bar-Shalom, Y. (1991). State estimation for systems with sojourn-time-dependent Markov model switching. IEEE Transactions on Automatic Control, 36(2), 238–243. https://doi.org/10.1109/9.67323
>
> Hu, Y., & Wager, S. (2022). Switchback experiments under geometric mixing. arXiv. https://arxiv.org/abs/2209.00197
>
> Johari, R., Peng, T., & Xing, W. (2025). Estimation of treatment effects under nonstationarity via truncated difference-in-Q's. arXiv. https://arxiv.org/abs/2506.05308

---

> > ### Comment · Reviewer_xsub · 2025-08-01
> >
> > Thank you for the detailed response. They addressed my concerns!

---

### Official Review · Reviewer_H7wU · 2025-07-07

**Rating:** 5
**Confidence:** 2

**Summary:**

This work introduces a simulation framework NS-Gym and benchmarking set for non-stationary MDPs. To the best of my understanding and research, this appears to be a novel addition to the literature (there are no standard benchmarks for non-stationary MDPs). Addressing this is a critical step in making non-stationary MDP algorithms practical and relevant.

**Additional Feedback:**

-

**Dataset Code Accessibility:**

Yes

**Ethical Considerations:**

No, there are no or only very minor ethics concerns

**Final Justification:**

My concerns regarding extensibility of the regret metrics have now been resolved, and I am confident that this work would contribute to a growing literature on NS-MDPs and would be a welcome addition to the field. I thank the authors for their detailed replies to my concerns. I am excited to see this benchmark used in future work in the field.

**Limitations Weaknesses:**

I find this work to be extremely useful, and so I don't have any major concerns.

However, the paper doesn't mention, and I assume it's not particularly hard to implement if it doesn't exist, but does the framework support (potentially) different kinds of regret measures? For instance, un-discounted regret and local/adaptive regret (see Refs. 1,2) which people in the online learning / control communities tend to care about?

I would be particularly interested in the above, and would be grateful if the authors could focus their reply towards an answer to the above: put simply, does the framework be made to extend to other regret measures (undiscounted, local, adaptive)? For posterity, can users define their own regret functions down the line? This would be very helpful for the field...

1) https://arxiv.org/abs/2008.05523
2) https://arxiv.org/pdf/2404.13009

**Strengths Contributions:**

- NS-Gym is introduced to deploy and benchmark non-stationary MDP algorithms using classic control experiments that are relevant in the community such as CartPole, Acrobat, FrozenLake, Pendulum, and few others. The authors have also already benchmarked many existing algorithms using it (such as DQN, alpha zero, etc.)

- This presents a unified way to approach experiments in this area, which are becoming more common as MDP algorithms are being shifted more towards the fully-online setting.

- The paper is extremely well-written (with a thorough background on the literature provided), and it provides a solid explanation for how to use the corresponding package in section 3.1, and details the formalism used to compute the Dynamic regret of the policy being learned.

- I particularly appreciate the fact that the authors have allowed the definitions to be as broad as possible (for instance, the agent definition is never concretely defined as it has been changing a lot in the recent literature).

---

> ### Author Rebuttal · Authors · 2025-07-31
>
> Thank you for your positive feedback and for highlighting the novelty and potential impact of our work.  We address your question about custom regret measures below.
>
> **Custom Regret Measures:**
>
> NS-Gym’s design is highly modular and extensible, seamlessly supporting custom regret measures. NS-Gym users can easily define custom evaluation modules and regret measures by inheriting from the package’s base `Evaluator` class. We demonstrate how users can define customer parameter update functions in the `tutorial.ipynb` file; the evaluation module can be extended in a similar manner. We will update the tutorial notebook with this example in the final version of the paper.

---

> > ### Comment · Reviewer_H7wU · 2025-08-01
> > **Reply to Authors**
> >
> > I thank the author for pointing this out.
> >
> > I would appreciate details on how these custom regret measures can be defined (for example, say local regret in Reference [2])? I'm curious because the comparator class may not be so easily accessible/computed, and this would be very impactful for future research in the field.
> >
> > Thanks,
> >
> > Reviewer

---

> > > ### Author Response · Authors · 2025-08-05
> > > **On Custom Regret Metrics**
> > >
> > > Thank you for your quick response and the insightful comment.
> > >
> > > Currently, the comparator class in our implementation is used to either compare two versions of the MDPs (e.g., through the PA-MCTS bound) or two trained policies (e.g., by computing the ensemble regret). To compute regret in the context of online policy optimization, such as the local regret defined in Lin et al. [1], we will need to extend the comparator class, as suggested by the reviewer. The time-varying nonlinear dynamics (unknown to the agent) can be implemented directly with NS-Gym, and the gradient of the loss function during learning can be computed and tracked using an automatic differentiation package. We will include an example of regret computation in the online policy optimization setting in the final version of the package and the tutorial.
> > >
> > >
> > > [1] Lin, Yiheng, et al. "Online policy optimization in unknown nonlinear systems." The Thirty Seventh Annual Conference on Learning Theory. PMLR, 2024.

---

> > > > ### Comment · Reviewer_H7wU · 2025-08-05
> > > > **Response to Authors**
> > > >
> > > > I thank the authors for their detailed reply. With this, all my concerns have been addressed and I maintain my positive recommendation of the paper.
> > > >
> > > > Thanks,
> > > >
> > > > Reviewer

---

### Note · Authors · 2025-08-13

We thank the reviewers and Area Chair for their constructive feedback and for recognizing the potential, importance, and structure of NS-Gym. We present a unified modeling framework and taxonomy for non-stationary Markov decision processes, drawing from decades of research in this domain. We also provide the first set of benchmarking results for non-stationary environments, along with open-source implementations of six state-of-the-art decision-making algorithms.

Our paper was well received by reviewers even before the rebuttal, and they agreed on the importance of this open-source toolkit and benchmark for the wider AI/ML community. The interaction with them and their constructive suggestions further strengthened the paper. We addressed all reviewer concerns, including extending the toolkit to extremely high-dimensional problems, refactoring the implementation of the CliffWalking environment to show a 67x improvement in running time, and presenting additional experimental results about parallel code execution through NS-Gym.

Once again, we would like to thank all the reviewers for their valuable time and constructive feedback.  We are also grateful to reviewer xzw8 for the productive dialogue that helped resolve an initial misunderstanding about their review.

---

### Decision · Program_Chairs · 2025-09-18

**Decision:**

Accept (poster)

**Comment:**

Summary

This paper describes an open source framework for non-stationary Markov processes.  It works in the limited domain of traditional problems like CartPole, Acrobat, FrozenLake, Pendulum, and few others.

Claims:

- Recent prior work on stationary Markov decision processes (MDP) use standard benchmark problems, e.g., by using the popular Gymnasium toolkit Towers et al. [2023], there are no standard problems or benchmarks for non-stationary MDPs.

The claims are true.

Strengths:
- Unique a unique "gym" to deploy and benchmark non-stationary MDP algorithms using classic control experiments that are relevant in the community such as CartPole, Acrobat, FrozenLake, Pendulum, and few others.
- Benchmarked on many existing algorithms such as DQN, alpha zero, etc.
- Presents a unified way to approach experiments in this area, which are becoming more common as MDP algorithms are being shifted more towards the fully-online setting.
-The paper is extremely well-written and it provides a solid explanation for how to use the corresponding package and details the formalism used to compute the Dynamic regret of the policy being learned.
- The authors have allowed the definitions to be as broad as possible (for instance, the agent definition is never concretely defined as it has been changing a lot in the recent literature).

Weaknesses:

- One criticism is that the problems are too low dimensional and it is not clear how this gym would scale to accommodate higher dimensional challenges that are being addressed by more modern machine learning methods.
- A lack of novel scientific insights beyond benchmarking
- Shallow experimental analysis


Scores: (from valid reviews) 5,5,4 where the reviewer rating "4" commented "Thank you for the detailed response. They addressed my concerns!" but did not raise their scrore to 5.  We had an additional review where the submitted review was generated using an LLM although and although the reviewer explained this by saying that the LLM use was for translation, the review introduced confusing artifacts, such as claiming that the paper was about "neuro-symbolic" computation and described spiking neural trains that are not in the paper.  We are choosing to ignore this original review.  The reviewer did submit a clarified review and did increast the score of that to positive, however given the initial confusion I am not fully counting this.